



# Environmental drivers and remote sensing proxies of post-fire thaw depth in Eastern Siberian larch forests

Lucas R. Diaz[1], Clement J. F. Delcourt[1], Moritz Langer[1,2], Michael M. Loranty[3], Brendan M. Rogers[4], Rebecca C. Scholten[1,5], Tatiana A. Shestakova[6,7], Anna C. Talucci[4], Jorien E. Vonk[1], Sonam Wangchuk[1], Sander Veraverbeke[1]

[1]Faculty of Science, Vrije Universiteit Amsterdam, de Boelelaan 1085, 1081 HV Amsterdam, The Netherlands
[2]Alfred Wegener Institute, Helmholtz Center for Polar and Marine Research, Telegrafenberg, 14473 Potsdam, Germany
[3]Department of Geography, Colgate University, Hamilton, NY 13346, United States of America
[4]Woodwell Climate Research Center, 149 Woods Hole Rd., Falmouth, MA 02540, United States of America
[5]Department of Earth System Science, University of California, Irvine, 3100 Croul Hall St., Irvine, CA 92697, United States of America
[6]Department of Agricultural and Forest Science and Engineering, University of Lleida, Av. Alcalde Rovira Roure 191, Lleida 25198, Spain
[7]Joint Research Unit CTFC–AGROTECNIO–CERCA, Av. Alcalde Rovira Roure 191, Lleida 25198, Spain

*Correspondence to*: Lucas R. Diaz (l.ribeiro.diaz@vu.nl)

**Abstract.** Boreal fire regimes are intensifying because of climate change and the northern parts of boreal forests are underlain by permafrost. Boreal fires combust vegetation and organic soils, which insulate permafrost, and as such deepen the seasonally thawed active layer and can lead to further carbon emissions to the atmosphere. Current understanding of the environmental drivers of post-fire thaw depth is limited but of critical importance. In addition, mapping thaw depth over fire scars may enable a better understanding of the spatial variability in post-fire responses of permafrost soils. We assessed the environmental drivers of post-fire thaw depth using field data from a fire scar in a larch-dominated forest in the continuous permafrost zone in Eastern Siberia. Particularly, summer thaw depth was deeper in burned (mean = 127.3 cm, standard deviation (sd) = 27.7 cm) than in unburned (98.1 cm, sd = 26.9 cm) landscapes one year after the fire, yet the effect of fire was modulated by landscape and vegetation characteristics. We found deeper thaw in well-drained landscape positions, in open larch forest often intermixed with Scots pine, and in high severity burns. The environmental drivers, site moisture, forest type and density, and fire severity explained 73.4 % of the measured thaw depth variability at the study sites. In addition, we evaluated the relationships between field-measured thaw depth and several remote sensing proxies. Albedo, the differenced Normalized Burn Ratio (dNBR), land surface temperature (LST), and pre-fire Normalized Difference Vegetation Index (NDVI) derived from Landsat 8 imagery together explained 66.3 % of the variability in field-measured thaw depth. Based on these remote sensing proxies and multiple linear regression analysis, we estimated thaw depth over the entire fire scar, and found that LST displayed particularly strong correlations with post-fire thaw depth ($r = 0.65$, $p < 0.01$). Our study reveals some of the governing processes of post-fire thaw depth development and shows the capability of Landsat imagery to estimate thaw depth at a landscape scale.



## 1 Introduction

Permafrost regions, covering Arctic and many boreal ecosystems, store approximately twice as much carbon as currently is in the atmosphere (Hugelius et al., 2014; Miner et al., 2022; Schuur et al., 2022). Permafrost ecosystems are rapidly changing because of climate warming (Natali et al., 2021; Schuur et al., 2022), and changes in fire regimes may accelerate permafrost thaw (Gibson et al., 2018; Chen et al., 2021a; Descals et al., 2022; Scholten et al., 2022). Fires are a major disturbance in permafrost landscapes (Walker et al., 2019; Holloway et al., 2020; Descals et al., 2022), influencing the carbon, energy and

water cycles (Li et al., 2021; Minsley et al., 2016; Larjavaara et al., 2017; Jin et al., 2021; Randerson et al., 2006; Liu et al., 2005).

Boreal fires combust parts of the aboveground biomass and organic soils (Walker et al., 2018, 2020b; Dieleman et al., 2020; Veraverbeke et al., 2021). When this insulating layer is consumed, charcoal is left on the surface reducing the albedo immediately after fire in the snow-free season (Yoshikawa et al., 2003; Rocha et al., 2012). This increases the absorption of

shortwave radiation and the accumulated heat results in ground warming. Moreover, the combustion of vegetation canopies increases the exposure of the ground to solar radiation and reduces the evaporative cooling effect (Yoshikawa et al., 2003; Shur and Jorgenson, 2007; Johnstone et al., 2010; Li et al., 2021; Liu et al., 2018; Zhao et al., 2021; French et al., 2016). The increased ground heat flux deepens the seasonally thawed active layer above the frozen ground (Holloway et al., 2020; Zhang et al., 2023, 2015). Active layer thickening can enhance soil respiration and consequent emissions of the greenhouse gases

carbon dioxide and methane for several decades (Köster et al., 2018; Veraverbeke et al., 2021; Genet et al., 2013; Taş et al., 2014; O'Donnell et al., 2011). Fires thus accelerate permafrost thaw, and although they are a natural component of boreal ecosystems (Kharuk et al., 2021), ongoing changes to the severity and extent of boreal fires (Turetsky et al., 2011b; Descals et al., 2022; Zheng et al., 2023) in combination with climate warming are projected to irreversibly change ice-rich permafrost environments (Gibson et al., 2018; Zhang et al., 2015; Holloway et al., 2020; Chen et al., 2021b).

Previous work has highlighted fires as an important driver of the thickening of the active layer (Gibson et al., 2018; Brown et al., 2016; Zhang et al., 2015; Jafarov et al., 2013). In addition, the active layer thaw depth is strongly controlled by vegetation type and density, organic soil content, soil texture, soil moisture content, topography and drainage (Park et al., 2013; Bai et al., 2018; Duguay et al., 2013; Loranty et al., 2018b; Fisher et al., 2016). Interactions between these diverse environmental variables lead to a large heterogeneity of the depth of the active layer in the landscape (Holloway et al., 2020; Liu et al., 2012;

Shiklomanov and Nelson, 2002). A higher canopy density reduces the amount of solar radiation reaching the ground, which in turn reduces the energy available for soil thawing (Fisher et al., 2016; Stuenzi et al., 2021a, b). In addition, vegetation type and density may alter soil moisture through evapotranspiration thereby affecting the thermal conditions of the permafrost active layer (Iwahana et al., 2005; Juszak et al., 2016; Fedorov et al., 2017; Marsh et al., 2010). Topography also influences the landscape's vulnerability to burning (Brown et al., 2016). For example, lowlands with limited drainage and wet soils tend

to burn with lower severity (Benscoter et al., 2011; Turetsky et al., 2011a; Dillon et al., 2011; Holloway et al., 2020). Thus, the active layer depth is governed by complex interactions between topography, vegetation, and fires (Brown et al., 2016).



However, quantitative understanding of these effects over space and time, as well as the strength of underlying drivers, are needed given the intensification of Arctic and boreal fire regimes (Natali et al., 2021; Gibson et al., 2018; Michaelides et al., 2019).

Remote sensing provides information on the ground surface and shallow soils, yet cannot directly measure the state of the subsurface. Nevertheless, remote sensing acquires information on various environmental drivers with known influence on active layer depth, or that are indicative of changes in it, such as vegetation status (Brown et al., 2016; Li et al., 2019), ground subsidence (Chen et al., 2020; Michaelides et al., 2019; Molan et al., 2018), land surface temperature (LST) (Park et al., 2016; Yi et al., 2018; Bai et al., 2018; Zorigt et al., 2020; Wen et al., 2022; Ran et al., 2022), surface albedo (Liu et al., 2018; Zhao

et al., 2021; Rogers et al., 2014) and fire severity (De Santis and Chuvieco, 2007; Lentile et al., 2006; Delcourt et al., 2021). LST has shown particular potential to map and model active layer depth over large areas given its relationship to the ground thermal regime (Batbaatar et al., 2020; Obu et al., 2019; Westermann et al., 2017; Hachem et al., 2012; Langer et al., 2010). Obu et al. (2021) produced an active layer thickness (ALT) dataset derived from a remotely sensed-driven model based on Moderate Resolution Imaging Spectroradiometer (MODIS) LST merged with reanalysis near-surface air temperature data. Liu

et al. (2024) used a machine learning approach with several MODIS products (including LST) as predictor variables to create 1 km gridded ALT. However, no studies have used Landsat thermal remote sensing data so far to assess and map thaw depth in a post-fire environment at a spatial resolution of 100 m and less, primarily because of the lower temporal resolution of Landsat data in comparison with for example MODIS. The higher spatial resolution from Landsat, however, may offer benefits to map and understand post-fire effects on permafrost soils with more spatial detail, thereby allowing comparisons with field

measurements.

In recent years, Eastern Siberian larch forests underlain by permafrost have experienced extreme fire activity (Xu et al., 2022; Talucci et al., 2022a, b; Scholten et al., 2022; Zheng et al., 2023). In this study, we investigated post-fire thaw depth in a fire scar in a larch-dominated forest underlain by continuous permafrost in Eastern Siberia using field and remote sensing data. Our objectives are twofold. First, we assessed which environmental drivers influence post-fire thaw depth. Second, we

investigated which remote sensing proxies relate to field-measured thaw depth. Using these relationships, we mapped post-fire thaw depth over the entire fire scar.

## 2 Materials and methods

### 2.1 Study area

Our study area is a fire scar near the Yert rural locality, Republic of Sakha (also known as Yakutia), Russia (Fig. 1). The fire

occurred between 30 June and 21 July 2018 and burned approximately 97 000 ha. The area is approximately 200 km West of Yakutsk, the capital of Yakutia. The mean annual air temperature in Yakutsk is -10.2 °C, with means of -42.6 °C in January and 18.7 °C in July, and the mean annual precipitation is 234 mm (Fedorov et al., 2017). The fire scar is in continuous permafrost terrain dominated by Cajander larch (*Larix cajanderi*), sometimes intermixed with the presence of Scots pine (*Pinus*





*sylvestris*) and silver birch (*Betula pendula*), and the tall shrubs alder (*Alnus spp.*) and willow species (*Salix spp.*). The surface
vegetation includes cowberry (*Vaccinium vitis-idaea*), bog blueberry (*Vaccinium uliginosum*), crowberry (*Empetrum nigrum*),
rhododendron (*Rhododendron dauricum*), dog-rose (*Rosa acicularis*), spirea shrub (*Spiraea spp.*), juniper (*Juniperus spp.*),
fireweed (*Epilobium angustifolium*), moss (as *Ceratodon purpureus* and *Aulacomnium palustre*), and lichens (Delcourt et al.,
2021). Cajander larch trees are deciduous needleleaf trees that are adapted to grow on permafrost terrain. They are
physiologically and morphologically adapted to withstand the low temperatures, short growing seasons, and cryogenic
processes (Abaimov, 2010; Berner et al., 2012). Cajander larch and Scots pine have evolved and adapted to frequent low
severity fires. The fire investigated in this study included a wide range of severities, from low severity surface fires to high
severity stand-replacing fires (Delcourt et al., 2021).

## 2.2 Field data

We collected field data in 13 burned and 7 unburned plots (Fig. 1) between July 30 and August 8 of 2019, approximately one
year after the fire (Delcourt et al., 2024). Plot selection aimed to cover a gradient in vegetation composition, fire severity, and
landscape position, within accessibility constraints. The plots were represented by quadrats of 30 m by 30 m to match the
spatial resolution of the Landsat Operational Land Imager (OLI) optical imagery. We aimed to sample plots in areas where
fire severity, vegetation, and landscape characteristics were relatively homogeneous (Delcourt et al., 2021).

For each plot, center coordinates and elevation were collected using a high-precision Global Positioning System handheld
device (Trimble Geo 7X, GeoExplorer) with 1 m horizontal and vertical accuracy. The geolocation was post-processed using
data from the closest reference station (Scripps Orbit and Permanent Array Center (SOPAC), Seismic Station Yakutsk),
resulting in decimeter accuracy. A clinometer was used to determine the slope. We also determined plot-level site moisture
classes following Johnstone et al. (2008). This approach assesses site moisture based on local topographic drainage thereby
accounting for soil texture and permafrost presence. The resulting ordinal scale consists of six site moisture classes ranging
between dry and wet (xeric, subxeric, subxeric to mesic, mesic, mesic to subhygric, and subhygric). This site moisture
classification has been used extensively for fire studies in boreal North America (Walker et al., 2020a).

We measured vegetation characteristics (stand age, basal area, vegetation density, and larch proportion) in a 30 m by 2 m
transect in the north-south direction of the plot and with the plot center as centroid. To characterize vegetation type and density,
we inventoried every live and standing dead tree along the transect. Particularly, we recorded tree species, measured diameter
at breast height (DBH; 1.3 m) and estimated basal area and stem density.

To characterize fire severity, we used the Geometrically structured Composite Burn Index (GeoCBI) protocol (De Santis and
Chuvieco, 2009). GeoCBI is a visual field-based fire severity index in a continuous numeric scale ranging from 0 to 3. The
GeoCBI estimates the cumulative impacts of fire over the vertical structure of forest stands and soil. Its field protocol is based
on a hierarchical and multi-layered sampling design, splitting the plot into five different strata: substrates (ground surface,
litter, duff); herbs, low shrubs, and trees less than 1 m; tall shrubs and trees of 1 to 5 m; intermediate trees of 5 to 20 m; trees
higher than 20 m. The strata are divided into subcategories that are evaluated independently according to various visible fire



effects. These subcategories for example include changes in the color and condition of the soil, consumption of fuels and the char height on tree boles. The subcategories are rated with decimal values between 0, meaning no fire effect, and 3, reflecting high fire severity. Then, the scores of each stratum are obtained by averaging the scores for all criteria and strata are weighted by their fraction of coverage to derive the plot-level GeoCBI value. Previous studies have used this index to assess fire severity in boreal forests (Rogers et al., 2014; Dieleman et al., 2020; Delcourt et al., 2021).

We estimated burn depth in the burned plots by measuring the vertical distance between the top of the residual soil layer and the uppermost adventitious root on larch trees, hereafter referred to as adventitious root height (ARH), following the methodology described in Delcourt et al. (2021). Adventitious roots are fine lateral roots that develop at upper soil horizons as trees grow, responding to unfavorable temperature and moisture conditions deeper in the soil (Rogers et al., 2014). These roots remain visible on the tree trunks several years after a fire (Kajimoto et al., 2003; Kajimoto, 2010). The position of adventitious roots in the soil column can be used to estimate the height of the pre-fire soil surface (Kasischke and Johnstone, 2005; Boby et al., 2010; Rogers et al., 2014). In unburned plots, we determined the location of the highest adventitious root beneath the moss layer as well as the soil organic layer depth (SOL) and derived a linear relationship between SOL and adventitious root height above the mineral soil. We used this relationship in burned plots to reconstruct pre-fire SOL from measurements of ARH and residual SOL depth. Burn depth was then estimated as the difference between reconstructed pre-fire SOL depth and residual SOL depth.

Thaw depth is defined as a measure of the seasonally unfrozen portion of the soil column (the active layer) that lies atop perennially frozen ground (permafrost) (Rocha et al., 2012). The thaw depth was measured using a standard frost probe (stainless-steel rod), which was inserted into the ground to the depth of resistance by the frozen ground. We performed measurements twice (with 1 m increments) every 7.5 m along the belt transect. In some measurements, we were uncertain whether the depth resistance stemmed from frozen ground or was caused by rocks. In case of uncertainty, we excluded this measurement from the analysis. Our approach resulted in a maximum of ten thaw depth measurements per plot, which were then averaged to derive the plot-level thaw depth. All our measurements were made between July 30 and August 8, 2019, and we therefore assume negligible seasonal influences on thaw depth between plots.

**2.3 Remote sensing data**

Landsat 8 Operational Land Imager (OLI) and Thermal Infrared Sensor (TIRS) Collection 2 Level-2 data were used in this study (Table 1). The OLI reflective and the TIRS thermal bands have spatial resolutions of 30 m and 100 m, respectively, but the latter are resampled and provided at 30 m for consistency purposes. We acquired post-fire images for July 23, 2019, the closest date to the field campaign. For the pre-fire imagery, we used a cloud-free image from July 7, 2016, near the anniversary date of the post-fire imagery. No cloud-free summer images were available from 2017. Two scenes were mosaicked to cover the entire fire perimeter for both pre- and post-fire imagery (Fig. 1). We used the Landsat data to calculate the pre-fire Normalized Difference Vegetation Index (NDVI), post-fire albedo, land surface temperature (LST) and the differenced Normalized Burn Ratio (dNBR).



We calculated the pre-fire NDVI as follows:

$$NDVI = \frac{\rho_5 - \rho_4}{\rho_5 + \rho_4} \tag{1}$$

where $\rho$ represents the reflectance of the Landsat 8 OLI bands 4 (0.64–0.67 µm) and 5 (0.85–0.88 µm). We included the pre-fire NDVI as a proxy of fuel availability.

To retrieve albedo ($\alpha$), we used the narrow-to-broadband conversion formula from Liang (2001) adapted for Landsat 8 as follows (Naegeli et al., 2017):

$$\alpha = 0.356\rho_2 + 0.130\rho_4 + 0.373\rho_5 + 0.085\rho_6 + 0.072\rho_7 - 0.0018 \tag{2}$$

where $\rho$ is the reflectance of the Landsat 8 OLI bands 2 (0.45–0.51 µm), 4, 5, 6 (1.57–1.65 µm), and 7 (2.11–2.29 µm).

The dNBR was obtained from the bitemporal difference of the Normalized Burn Ratio (NBR) (Eq. (3)) between pre- and post-fire scenes (Eq. (4)) (García and Caselles, 1991; Key and Benson, 2006; Epting et al., 2005):

$$NBR = \frac{\rho_5 - \rho_7}{\rho_5 + \rho_7} \tag{3}$$

$$dNBR = NBR_{pre-fire} - NBR_{post-fire} \tag{4}$$

The dNBR is an often-used fire severity index and theoretically can range between -2 and 2. In practice, however, most pixel values range between 0 and 1. A dNBR value of zero denotes an unburned pixel, and increasing dNBR values represent higher fire severity (Key and Benson, 2006; Allen and Sorbel, 2008).

For the LST retrieval, we opted for an approach based on the direct inverse solution of the radiative transfer equation (RTE) (Price, 1983). This is a physically based method that is often used with a single thermal band (Jiménez-Muñoz et al., 2009). Inverting and simplifying the RTE to a thermal band center at wavelength $\lambda$, the LST can be calculated as:

$$LST = \frac{K_2}{ln\left(\frac{K_1}{\frac{L_\lambda^{sen} - L_\lambda^\uparrow - \tau_\lambda(1-\varepsilon_\lambda)L_\lambda^\downarrow}{\tau_\lambda \varepsilon_\lambda}} + 1\right)} \tag{5}$$

where $L_\lambda^{sen}$ refers to the at-sensor spectral radiance of the corresponding thermal band. In this study, we used the Landsat 8 TIRS band 10 (10.60–11.19 µm). $\varepsilon_\lambda$ represents the land surface emissivity (LSE). $K_1$ and $K_2$ are calibration constants for band 10 of the Landsat 8 TIRS (Ihlen and Zanter, 2019). $L_\lambda^\downarrow$ and $L_\lambda^\uparrow$ are the downwelling and upwelling atmospheric radiances and $\tau_\lambda$ is the atmospheric transmittance. These atmospheric parameters were derived from the Goddard Earth Observing System, Version 5 (GEOS-5) Forward Processing for Instrument Teams (FP-IT) data, which are available as atmospheric auxiliary data of the Landsat 8 products (Engebretson, 2020; Sayler and Glynn, 2022).

LSE information is essential for the LST retrieval. LSE determines the efficiency of the surface to convert heat energy into radiant energy. We adopted the NDVI threshold method (NDVI$^{THM}$) of Sobrino et al. (2004) to estimate LSE, but we adapted the method to the specific characteristics of a fire scar. For that, we first computed the NDVI and the fractional vegetation cover ($P_V$) (Carlson and Ripley, 1997):



$$P_V = \left(\frac{NDVI - NDVI_{min}}{NDVI_{max} - NDVI_{min}}\right)^2 \tag{6}$$

where $NDVI_{min} = 0.2$ and $NDVI_{max} = 0.5$ (Sobrino and Raissouni, 2000; Sobrino et al., 2004, 2008). Then, we estimated LSE as follows. The spectral notation ($\lambda$) is omitted, since here a single TIRS band was used.

$$\varepsilon = \begin{cases} 0.962 & 0.0 \leq NDVI < 0.2 \\ 0.990P_V + 0.962(1 - P_V) + d\varepsilon & 0.2 \leq NDVI \leq 0.5 \\ 0.990 & NDVI > 0.5 \\ 0.993 & NDVI < 0.0 \end{cases} \tag{7}$$

$$d\varepsilon = (1 - 0.962)0.990F'(1 - P_V) \tag{8}$$

in this approach, pixels with NDVI values between 0 and 0.2 were considered as non-vegetated ($P_V = 0$). Since our study
focuses on a fire, a LSE value of 0.962 was assigned to these pixels. This value was derived from charcoal spectra from Veraverbeke et al. (2012, 2014) integrated over the Landsat 8 TIRS band 10 spectral response function. If the pixel had an NDVI value between 0.2 and 0.5, it was a mixture of charcoal and vegetation and LSE depended on the $P_V$ value calculated according to Eq. (7). $d\varepsilon$ accounts for the cavity effect due to surface roughness (i.e., $d\varepsilon = 0$ for flat surfaces) and was approximated by Eq. (8), with an assumed shape factor $F'$ of 0.55 following Sobrino et al. (1990). Pixels with NDVI values
higher than 0.5 were fully vegetated ($P_V = 1$) and we assumed an emissivity of 0.990. Finally, when the pixel had a negative NDVI value corresponding to water surface, an LSE value of 0.993 was assigned (Vanhellemont, 2020).

We calculated the average pixel values for all remote sensing metrics within a three-by-three pixel window centered on a given plot's centroid coordinates, which minimized the effects of potential satellite misregistration when comparing these remote sensing metrics with field data (Ahern et al., 1991). We also compared NDVI, albedo, dNBR and LST values of burned pixels
from within the fire perimeter with values of these variables of unburned pixels within a 2 km buffer from the fire perimeter.

## 2.4 Statistical approach

Our statistical analysis consisted of two major components. First, we assessed the environmental drivers of post-fire thaw depth using our field observations. Second, we evaluated how remote sensing variables can be used to predict field-measured thaw depth. For both analyses, we used an ordinary least squares (OLS) multiple linear regression (MLR) model with thaw
depth as the response variable using the "statsmodels" Python module (Seabold and Perktold, 2010). We hypothesized that post-fire thaw depth relates to variables associated with topographic position (site moisture, slope, and elevation), vegetation (stand age, basal area, vegetation density, and larch proportion), and fire severity (GeoCBI and burn depth). The selection of the environmental variables was based on the individual correlations of these variables with thaw depth and independence between predictor variables. As such, variables with an absolute Pearson correlation lower than 0.4 were excluded from the
MLR. When predictor variables were mutually correlated with an absolute Pearson correlation higher than 0.7, then only the predictor variable with the highest absolute correlation with thaw depth was retained in the MLR. Finally, we developed a MLR model using the remote sensing metrics (post-fire albedo, dNBR, LST, and pre-fire NDVI) as predictor variables to





assess their capacity as proxies for post-fire thaw depth. The resultant obtained MLR was then spatially extrapolated to derive a continuous map of post-fire thaw depth over the study area.

In addition to these statistical analyses, which included all field plots, we also focused separately on two burned-unburned plot pairs. While our study area captured a large variability in landscape position, vegetation characteristics, and fire severity, these plots had the advantage of being spatially adjacent with a road acting as the fire barrier. Because of this, the landscape and vegetation characteristics of these plots were very similar. As a result, these plot pairs represent interesting study cases in which the fire effect on thaw depth can be unequivocally separated from the landscape and vegetation influences.

## 3 Results

### 3.1 Environmental drivers of post-fire thaw depth

On average, summer thaw depth was deeper in burned (mean = 127.3 cm, standard deviation (sd) = 27.7 cm) than in unburned (98.1 cm, sd = 26.9 cm) plots (Fig. 2). Based on their individual correlations with thaw depth (Fig. A1), we retained site moisture, larch proportion, vegetation density, and GeoCBI as predictor variables for the MLR. These predictor variables 235 correlated reasonably strongly with thaw depth (Fig. 3), yet did not correlate strongly with each other and thus represent an independent set of predictors. Thaw depth was deeper in well-drained landscape positions (Fig. 3a), in mixed forests which included tree species other than Cajander larch (Fig. 3b), open forests (Fig. 3c) and in high severity burns (Fig. 3d). The multiple linear regression results indicated that the selected environmental drivers together explained 73.4 % of the thaw depth variability (Table 2). Thaw depth was also deeper in uplands (Figs. A2a and A2c) and in mature forests (Fig. A2d).

### 3.2 Remote sensing proxies of thaw depth

Within the fire perimeter, there was considerable spatial variability in albedo, dNBR, LST, and pre-fire NDVI (Figs. 4 and 5). Albedo values inside the fire perimeter had a mean of 0.10 (sd = 0.02), compared to mean of 0.12 (sd = 0.02) outside the perimeter (Figs. 4a and 5a). The dNBR values from our 13 burned field plots ranged between 0.34 and 0.85, with a mean of 0.57, while for the entire fire perimeter the mean was 0.41 (sd = 0.27, Fig. 4b and 5b). In addition, LST values inside the fire 245 scar were on average 5.24 K higher than the LST values outside the fire scar (Figs. 4c and 5c). However, for some parts of the fire scar, LST values exceeded the surrounding unburned pixels by approximately 20 K. For the pre-fire NDVI, the distributions of values inside and outside the fire perimeter were similar (Figs. 4d and 5d) with a mean NDVI value of 0.78 (sd = 0.08) inside the perimeter and and a mean NDVI value of 0.79 (sd = 0.07) outside the perimeter.

Thaw tended to be deeper in the plots with lower summer albedo (Fig. 6a), higher dNBR, (Fig. 6b), higher LST (Fig. 6c) and 250 lower pre-fire NDVI (Fig. 6d). The correlation with LST was the strongest, while the correlations between thaw depth and dNBR, albedo and pre-fire NDVI were not statistically significant at $p < 0.05$.





Taken together, the remote sensing proxies explained 66.3 % of field-measured thaw depth variability. The strongest remote sensing predictor was LST, which captured 42.9 % of the variability in field-measured thaw depth (Table 3). LST also captured variation related to site moisture, larch proportion, and GeoCBI (Figs. 7 and B1).

The MLR model for thaw depth based on remote sensing proxies enabled spatially continuous thaw depth estimates over the study area, including uncertainty estimates (Fig. 8). Post-fire thaw depth varied largely within the fire scar varied, for example between 83.49 cm (5th percentile) and 226.42 cm (95th percentile), with a mean of 139.72 cm (sd = 43.76 cm). In contrast, for the pixels outside the fire perimeter, the estimated thaw depth varied between 66.59 cm (5th percentile) and 237.99 cm (95th percentile), with a mean of 125.08 cm (sd = 52.15 cm) (Fig. 8c). On average, burned areas thus experienced 14.64 cm
deeper summer thaw depth one year after the fire than unburned areas.

**3.3 Differences between neighboring burned and unburned plots**

Our field dataset contained two pairs of burned-unburned plots, which were separated by a road (Fig. 9). The landscape and vegetation characteristics were very similar between adjacent burned and unburned plots and represented mature larch-dominated forest in a subxeric to mesic environment (Table 4). The burned plots experienced medium to high fire severity
with GeoCBI values of approximately 2.5 and burn depth of 10.5 cm. The thaw depth in the two burned plots was 53 cm and 73 cm deeper compared to the paired unburned control plots. In addition, LST values in the burned plots were between 3 and 5 K higher than in the unburned plots, and the burned plots experienced a clear drop in albedo.

**4 Discussion**

**4.1 Environmental drivers of post-fire thaw depth**

Fire accelerates permafrost active layer thickening (Gibson et al., 2018). However, this process is also modulated by topographic and vegetation characteristics (Fisher et al., 2016; Brown et al., 2015, 2016). We found that well-drained terrain positions generally experience deeper thaw. Previous studies have reported contrasting findings concerning moisture effects. For example, Fisher et al. (2016) found deeper thaw with increased soil surface moisture. Other studies, in contrast, found thinner active layers for wet soils (Clayton et al., 2021; Zhang et al., 2005) because these wet soils require more energy for all
ice to melt (i.e., latent heat of fusion) resulting in shallower active layers (Clayton et al., 2021).We found that forest density was a thaw-limiting factor, in agreement with Fisher et al. (2016), who attributed canopy shading and evapotranspiration as mechanisms to explain this relationship between forest density and active layer depth. With denser canopies, less solar radiation can reach the ground, be absorbed and converted into heat, resulting in a shallower thaw (Fedorov et al., 2017; Juszak et al., 2014, 2016; Iwahana et al., 2005). Denser vegetation may also transpire more, thereby reducing the thermal conductivity in the soil (Fisher et al., 2016; Iwahana et al., 2005). Moreover, the presence of Cajander larch trees is closely coupled with



permafrost (Zhang et al., 2011; Herzschuh, 2020). Our results reflect this vegetation-permafrost interaction, where plots with fewer larch trees showed a more pronounced thaw.

Fire results in a rise in surface and soil temperature, which consequently increases thaw depth (Li et al., 2019; Nossov et al., 2013; Jiang et al., 2015; Smith et al., 2015; Gibson et al., 2018). Our results are in agreement with the overall pattern of fast
active layer thickening in the years immediately after the fire (Holloway et al., 2020). We found a mean difference of about 30 cm in field-measured summer thaw depth between burned and unburned plots one year after the fire, and our study thereby also fills a data gap for the effects of fires on active layer thickening in Siberia (Ponomarev et al., 2020; Petrov et al., 2022). Our case studies of two burned-unburned plot pairs divided by a fire barrier enabled us to separate the impact of fire on the active layer thaw from topographic and vegetation influences.

Increasing fire severity tends to increase thaw depth (Jafarov et al., 2013; Li et al., 2019; Jiang et al., 2015; Holloway et al., 2020; Alexander et al., 2018). The positive correlation between GeoCBI and thaw depth found in this study is consistent with these previous results. In addition, fire severity is often influenced by vegetation and topographic conditions. In our study for example, the well-drained plots are on upland landscape positions with deeper thaw depths, while lowland areas are usually wetter with limited drainage and shallow thaw (Figs. A1 and A2). These wet lowland ecosystems also tend to burn with lower
severity  (Benscoter et al., 2011; Turetsky et al., 2011a; Dillon et al., 2011; Holloway et al., 2020). The thaw measurements in burned plots, as measured in the field, are conservative estimates as we did not account for the combustion of the organic layer in these measurements. We found a mean burn depth of 10.2 cm (sd = 1.1 cm, Fig. A2f) in our burned plots. As a consequence, the thaw depth in reference to the pre-fire ground level is even deeper than compared to the post-fire ground level. This further elucidates the importance of fire severity, or burn depth, as a driver of post-fire permafrost thaw.

**4.2 Remote sensing proxies of thaw depth**

Charcoal residues after a fire are dark in color, decreasing the summer albedo in the fire scar. More solar radiation is absorbed by the darker surface, leading to a thickening of the thawed layer. Liu et al. (2018) and Zhao et al. (2021) assessed post-fire summer albedo change in the boreal forests of Eastern Siberia and North America. They reported small immediate (1 year after the fire) albedo declines, a maximum decline of 0.02 in Siberia and a mean decline of 0.01 in North America. We found
a mean drop of 0.02, equaling a relative decrease of 17 %, in summer albedo due to fire. The studies by Liu et al. (2018) and Zhao et al. (2021) used MODIS products with coarser spatial resolution, which may partly explain the differences with our study which used 30 m Landsat data. While the individual relationship between albedo and thaw depth was not strong, albedo may still be a useful metric to assess fire influences on permafrost soils, in part because albedo is found to be a good proxy for aboveground fire severity (Fig. B1; Rogers et al., 2014). In line with previous studies (Rogers et al., 2014; Allen and Sorbel,
2008; French et al., 2008; Veraverbeke et al., 2015; Delcourt et al., 2021), the dNBR was also strongly correlated with GeoCBI and as a result related reasonably strong with thaw depth too ($r = 0.41$, $p = 0.07$). Somewhat surprising, pre-fire NDVI did not perform well as a proxy for fuel density as it showed a low correlation with the field-measured pre-fire vegetation density (Fig.



B1). This may be because the NDVI in Siberian larch forest is strongly influenced by understory vegetation, thereby confounding its value as proxy for forest density (Loranty et al., 2018a; Bendavid et al., 2023).

We found that LST was the strongest remotely sensed predictor of post-fire thaw depth. This can be explained by the fact that it captures variations in topography, vegetation, and fire severity, which all showed important influences on post-fire thaw depth. LST has been used before to model ALT (Obu et al., 2019; Wen et al., 2022; Ran et al., 2022) and to study the impacts of fire on biophysical processes (Liu et al., 2018; Zhao et al., 2021). Although these studies used LST with a different approach, using time series at a lower spatial resolution, we have built upon these studies by using LST in a predictive model for
estimating post-fire thaw depth at landscape scale.

We demonstrated the potential of Landsat imagery to predict post-fire thaw depth, resulting in a spatially continuous map of post-fire thaw depth at landscape scale. The MRL model using the Landsat derived proxies was able to explain 66.3 % of the field-measured thaw depth variability. This performance is only slightly lower than the 73.4 % explained variability based on only field data. Despite this achievement, some limitations are worth mentioning. The Landsat 8 TIRS images have a native
spatial resolution of 100 m, but are distributed at 30 m after a non-reversible cubic convolution resampling. Even with data collected at lower native spatial resolution than the optical bands, LST was still our best remote sensing proxy of post-fire thaw depth. The availability of LST data at a higher spatial resolution would therefore likely be beneficial for the model performance. In addition, our field dataset was somewhat limited in size. Because of this, the MRL models should be seen as first-order statistical models of the relationships between field-measured and spaceborne environmental variables with thaw
depth. As more measurements of post-fire permafrost thaw may become available over time (Loranty et al., 2021), more advanced statistical approaches, including machine learning techniques, could be explored to better understand and map post-fire permafrost thaw over larger spatiotemporal scales.

We used a snapshot of LST that temporally overlapped with the field data collection, using only one image. Alternatively, one could use all available images within the growing season and gap fill with e.g. coarse spatial but high temporal resolution
MODIS LST imagery. It might thereby be possible to apply the concept of growing or thawing degree days, which is often used to approximate thaw depth evolution and ALT (Romanovsky and Osterkamp, 1997; Hinkel and Nelson, 2003; Streletskiy et al., 2008, 2015). In addition, we measured and estimated thaw depth in the middle of the growing season. However, the most common variable of interest for intercomparison is ALT, i.e. the maximum depth at the end of the growing season (Zhang et al., 2021; Michaelides et al., 2019; WMO, 2016). The rate at which the active layer deepens decreases towards the end of
the growing season. The thaw depth in August has been found to be approximately 6 % less than the ALT (Clayton et al., 2021; Hinkel et al., 2001; Michaelides et al., 2019; Zhang and Stamnes, 1998; Boike et al., 1998). Streletskiy et al. (2008) have found that 95-99 % of maximum thaw propagation is achieved by mid-August and that landscape-specific thaw depth patterns are related with landform elements that show spatial regularity at the landscape scale. We thus assumed that the spatial variability of the mid-season thaw depth is a reliable indicator of the spatial variability of ALT at the end of the growing season
in the study region.



For future research, in addition to optical and thermal data, Synthetic Aperture Radar data (SAR) datasets can be used to study spatiotemporal variations of fire effects on permafrost landscapes. SAR data enables the measurement of various physical properties of fire scars, including surface roughness and near-surface soil moisture. Moreover, the interferometric SAR (InSAR) technique allows precise measurements of ground movement, with a millimeter level of accuracy (Strozzi et al.,
2018). The effectiveness of this approach has been demonstrated in previous studies conducted in permafrost environments in Alaska and Siberia (Michaelides et al., 2019; Yanagiya and Furuya, 2020; Yanagiya et al., 2023). Future work could integrate fire-induced permafrost subsidence from InSAR with optical and thermal data to predict post-fire thaw depth based on multi-sensor satellite products.

## 5 Conclusions

Boreal fires combust parts or all of the aboveground vegetation and soil organic layer. The resulting loss of insulation of the permafrost soil leads to an increased ground heat flux, promoting permafrost thaw. With the escalating extent and severity of fires in Siberia, there is a growing need to better understand the processes that govern fire-induced permafrost thaw, including spatial estimations of thaw depth. Here, we evaluated the impact of fire, landscape, and vegetation characteristics on post-fire thaw depth. In addition, we assessed the capacity of remote sensing proxies to estimate post-fire thaw depth at a landscape
scale. Using field data from a recent fire scar in Eastern Siberia larch-dominated forests, we confirmed that increasing fire severity indeed induces deeper thaw. This process was, however, mediated by landscape and vegetation characteristics. Thaw depth was deeper in well-drained uplands, characterized by open and mature forests, which included mixed stands of Cajander larch and Scots pine. The environmental drivers, site moisture, forest type and density, and fire severity together explained 73.4 % of the thaw depth variability. We explored the use of Landsat 8-derived metrics as proxies for post-fire thaw depth.
Multiple linear regression model ingesting albedo, dNBR, LST, and pre-fire NDVI explained 66.3 % of the field-measured variability in thaw depth. The correlation between LST and thaw depth was particularly strong, highlighting the potential of LST as a proxy for active layer thaw depth. We estimated post-fire thaw depth at a landscape scale by spatially extrapolating the regression model of the Landsat reflective and thermal data. Future works should further investigate the consistency of our findings across space and time to better understand the impact of fire on permafrost ecosystems across the Arctic-boreal region.





**Appendix A**

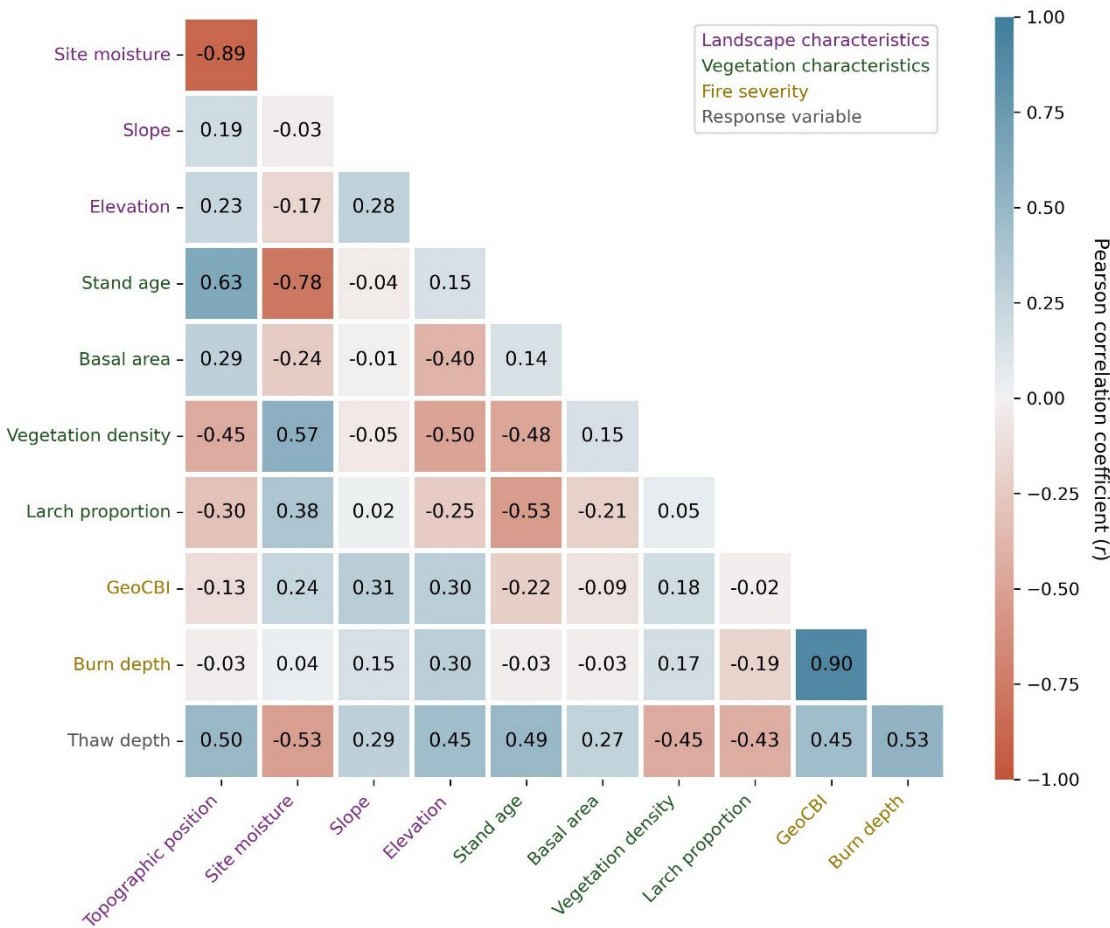

**Figure A1: Correlation matrix between field-measured environmental variables representing landscape, vegetation and fire severity characteristics, and thaw depth. GeoCBI is the Geometrically structured Composite Burn Index.**




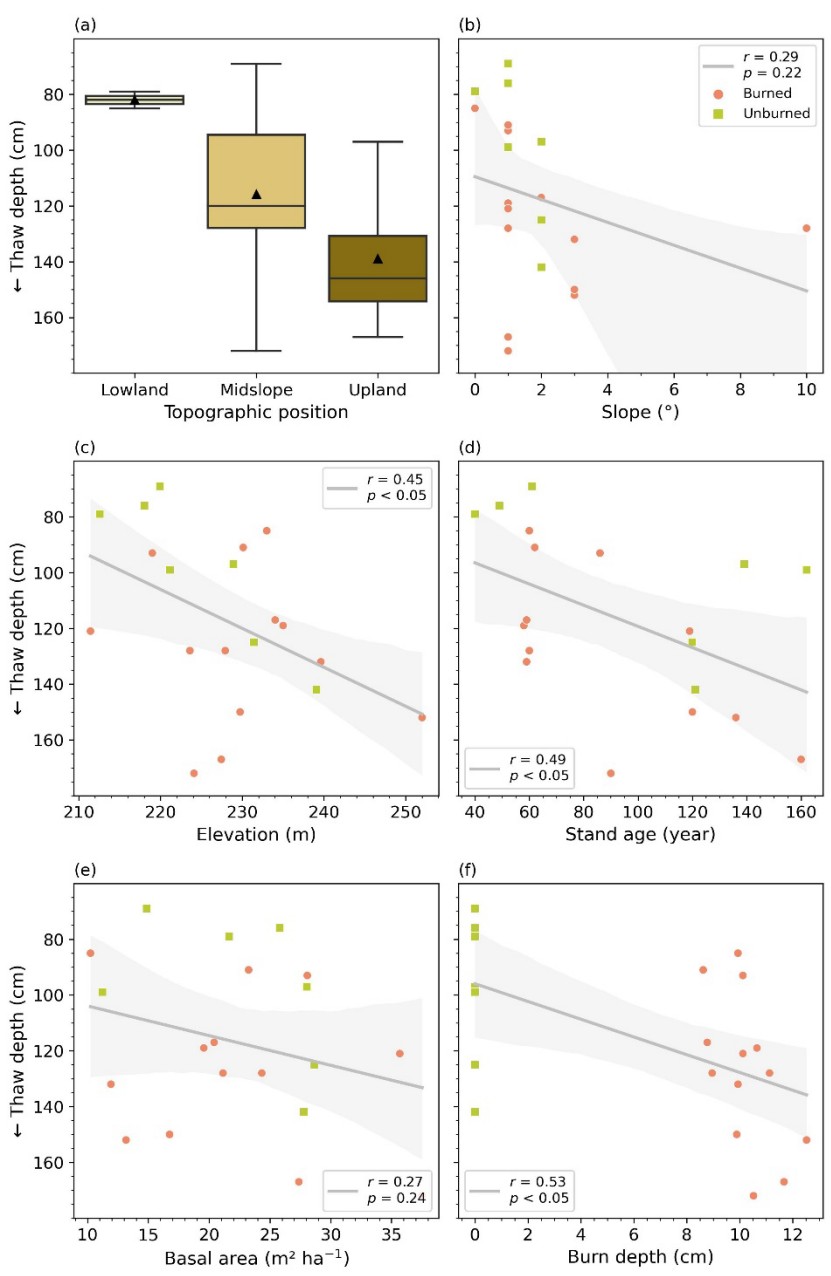

**Figure A2: Relationships between (a) topographic position, (b) slope, (c) elevation, (d) stand age, (e) basal area, (f) burn depth and thaw depth.**





**Appendix B**

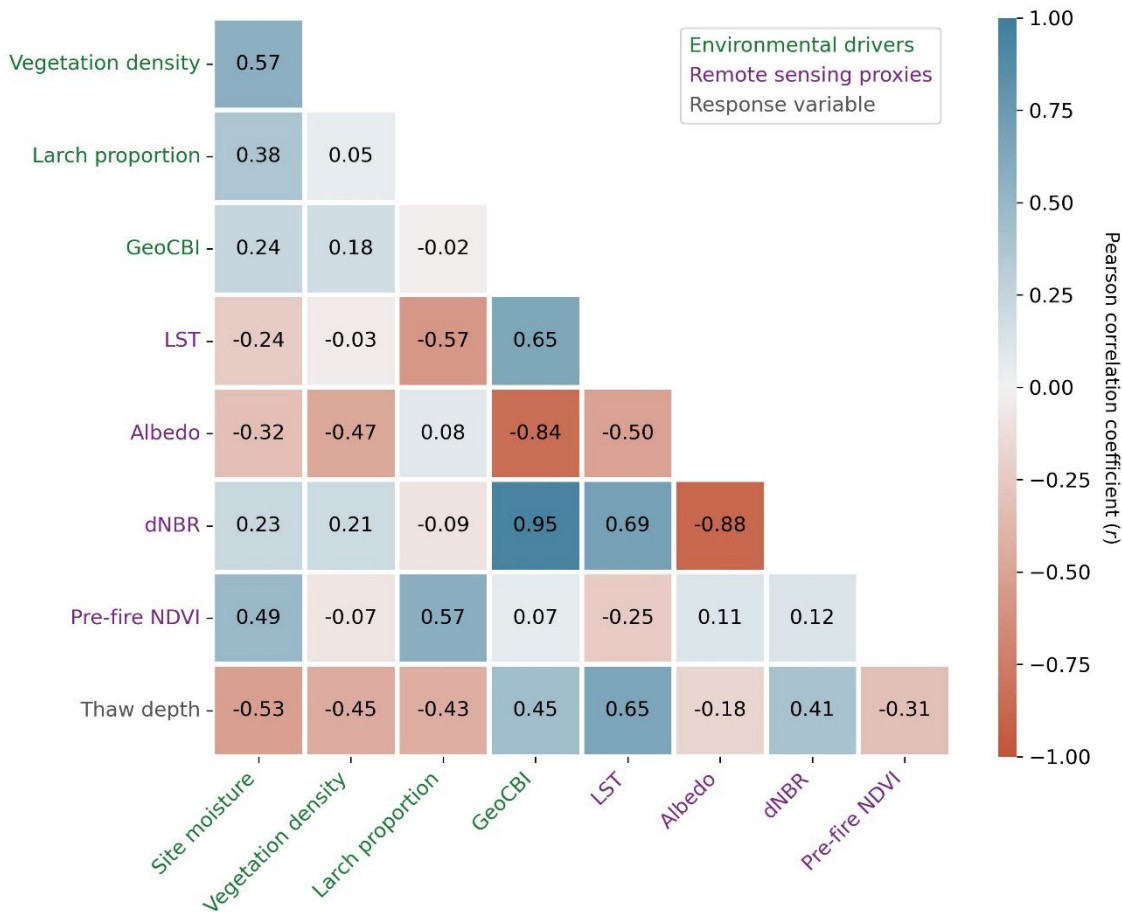

**Figure B1: Correlation matrix of selected environmental drivers, remote sensing proxies, and thaw depth. GeoCBI is the Geometrically structured Composite Burn Index. LST is the land surface temperature. dNBR refers to the differenced Normalized Burn Ratio and NDVI to the Normalized Difference Vegetation Index.**

**Code and data availability**

Field data can be accessed at: https://doi.org/10.5281/zenodo.10840088 (Delcourt et al., 2024). Landsat 8 Operational Land Imager (OLI) and Thermal Infrared Sensor (TIRS) data are publicly available and a courtesy of the U.S. Geological Survey (https://doi.org/10.5066/P9OGBGM6). The code with the statistical methods used to analyze the data can be obtained from the corresponding author upon request.



**Author contribution**

LRD and SV conceptualized the research and developed the methodology. CJFD and SV organized the field campaign. CJFD, BMR, RCS, TAS, and SV collected the field data. LRD performed the formal analysis, investigation, and visualization. LRD wrote and prepared the original draft with inputs from SV and SW. CJFD, ML, MML, BMR, RCS, TAS, ACT, JEV, SW, and SV reviewed and edited the manuscript. SV acquired the funding, administrated the project, and supervised the research.

**Competing interests**

The authors declare that they have no conflict of interest.

**Acknowledgments**

We thank Trofim C. Maximov and Roman E. Petrov for logistical and field support. We are grateful to Frédéric van Vessem, who performed a first exploratory analysis on the thaw depth dataset as part of his MSc thesis.

**Financial support**

This work was supported by the Dutch Research Council (NWO) through a Vidi grant (grant no. 016.Vidi.189.070) and by the European Research Council (ERC) through a Consolidator grant under the European Union's Horizon 2020 research and innovation programme (grant no. 101000987), both awarded to SV. TAS acknowledges the support received by the Beatriu de Pinòs Programme and the Ministry of Research and Universities of the Government of Catalonia (2020 BP 00126).

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





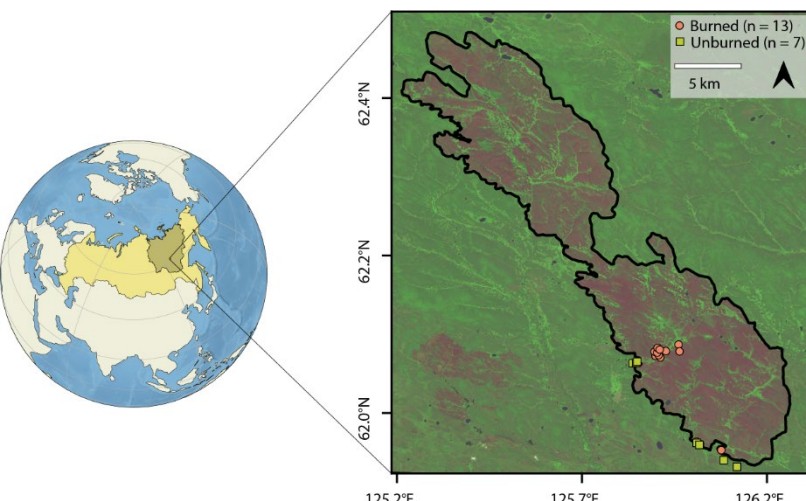

**Figure 1: Map of the study area in the Republic of Sakha, Russia. The fire scar is shown in the red-brown colors in a Landsat 8**
**Operational Land Imager false color composite (RGB–754) from 23 July 2019. The locations of burned and unburned field plots are**
**shown on the map.**

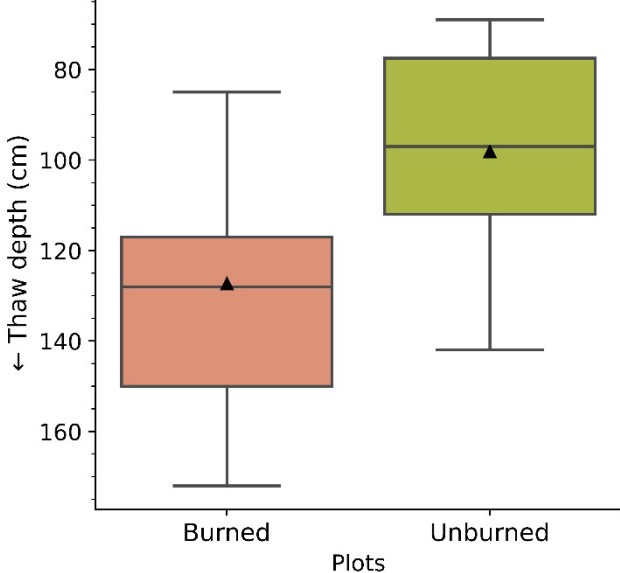

**Figure 2: One year post-fire summer thaw depth was deeper in burned plots compared to unburned plots. Each box ranges from**
**the first to the third quartile. Whiskers extend to points that lie within 1.5 times the interquartile range. The median is indicated by**
**the horizontal line and the mean by the black triangle.**



**Figure 3: Relationships between field-measured thaw depth and (a) site moisture, (b) larch proportion, (c) vegetation density, (d)**
**Geometrically structured Composite Burn Index.**



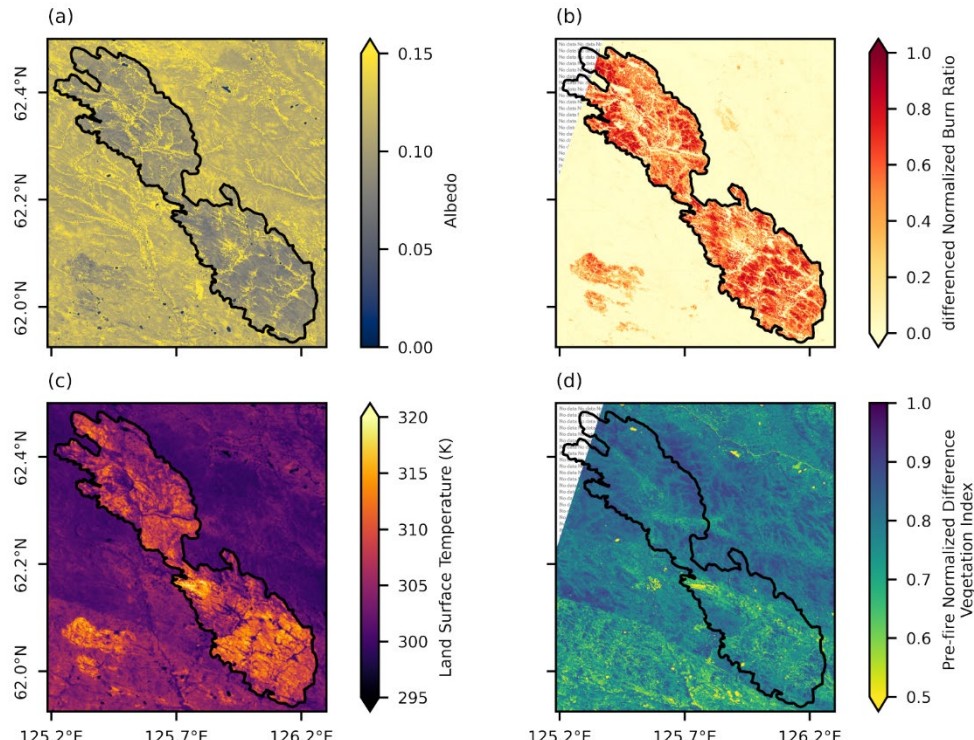

**Figure 4: Maps showing (a) albedo, (b) differenced Normalized Burn Ratio, (c) land surface temperature, and (c) pre-fire Normalized Difference Vegetation Index derived from Landsat 8 imagery.**



**Figure 5: Relative frequency distributions of remote sensing proxies derived from Landsat 8 imagery inside and outside the fire perimeters:(a) albedo, (b) differenced Normalized Burn Ratio, (c) land surface temperature, and (d) pre-fire Normalized Difference Vegetation Index. The pixels outside the fire perimeter were within 2 km from the fire perimeter.**








**Figure 6: Scatter plots and linear regression lines between field-measured thaw depth and remotely sensed (a) albedo, (b) differenced Normalized Burn Ratio, (c) land surface temperature, (d) pre-fire Normalized Difference Vegetation Index.**






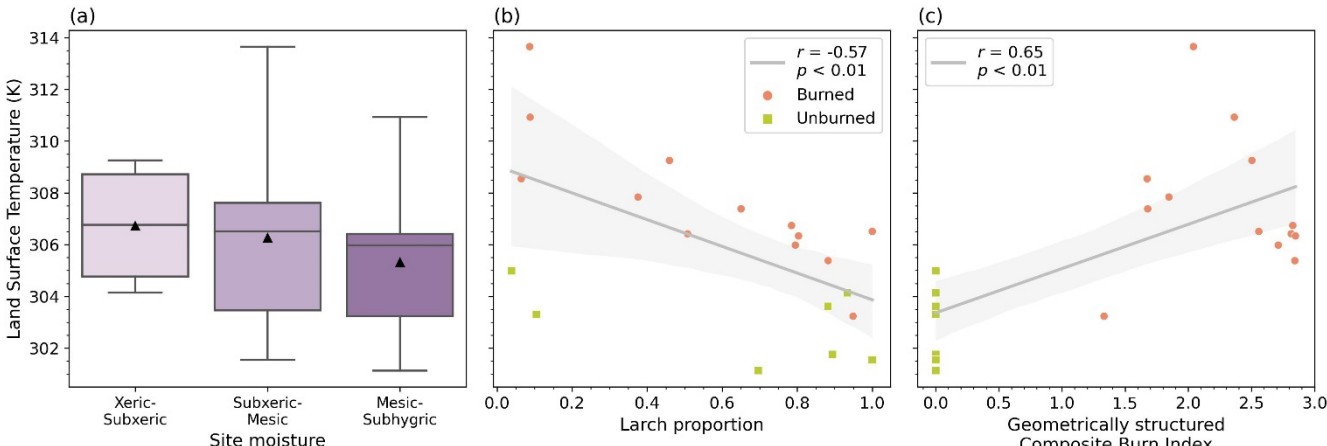

**Figure 7: Relationships between remotely sensed land surface temperature (LST) and field-measured (a) site moisture, (b) larch proportion, (c) Geometrically structured Composite Burn Index.**

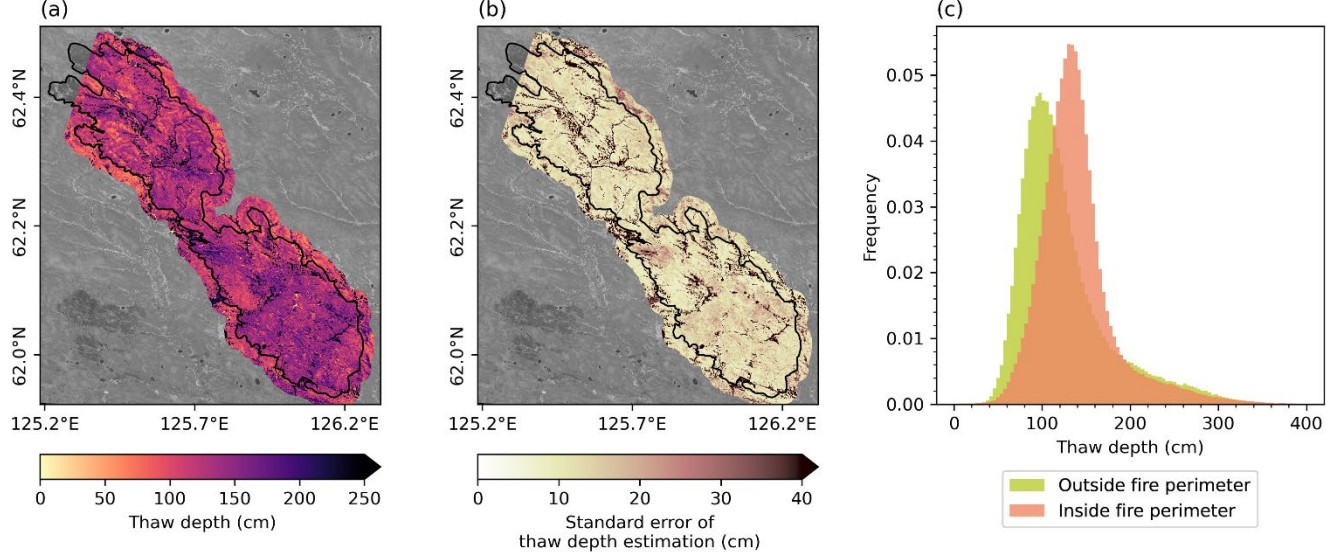


**Figure 8: Maps of (a) estimated thaw depth, (b) its standard error, and the (c) relative frequency distribution for the pixels inside and outside the fire perimeter. The area outside the fire scar consider a 2 km buffer from the fire perimeter.**



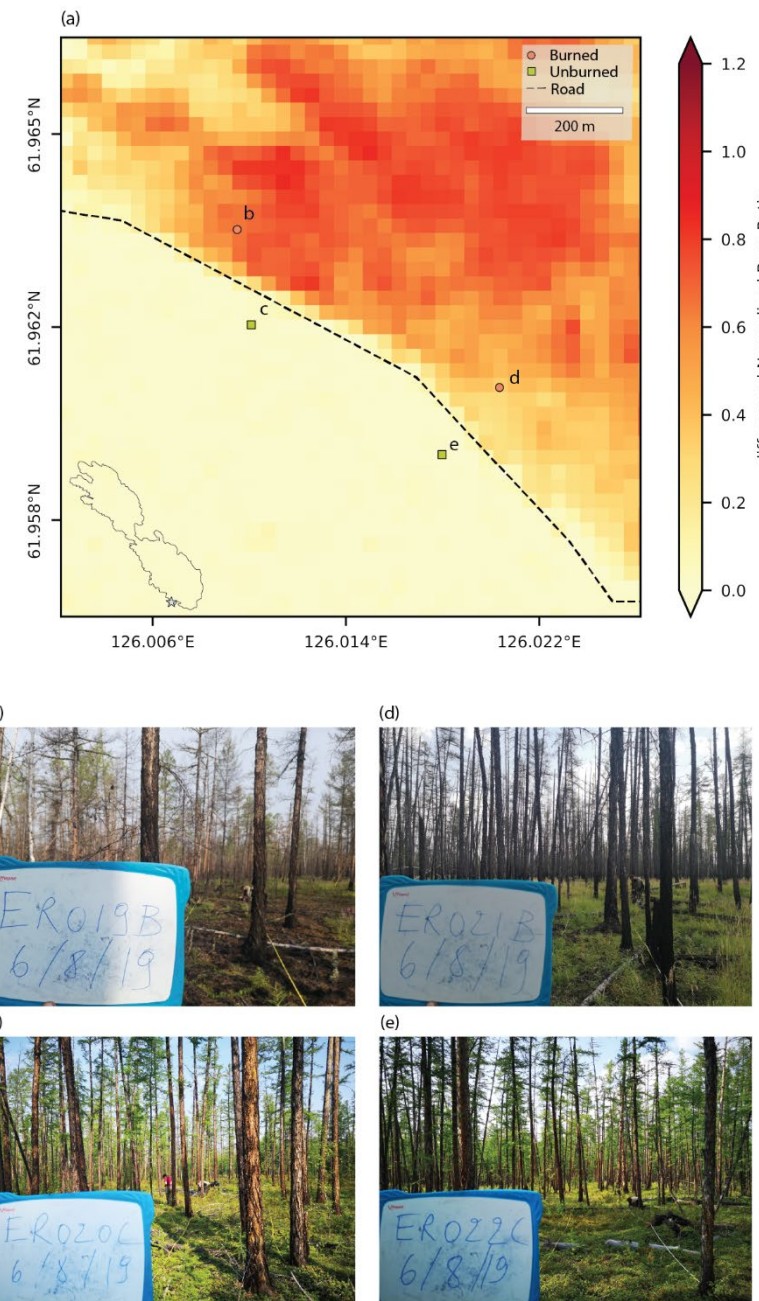

**Figure 9: Location of the two pairs of burned-unburned plots in the study area (a), including field photographs, which contain the plot identification labels (b-e).**





**Table 1: Metadata of Landsat 8 imagery used in this study.**

| | Scene ID | Acquisition date | Path | Row |
|---|---|---|---|---|
| **Pre-fire** | LC81230162016189LGN01 | 07 July 2016 | 123 | 16 |
| | LC81230172016189LGN01 | 07 July 2016 | 123 | 17 |
| **Post-fire** | LC81240162019204LGN00 | 23 July 2019 | 124 | 16 |
| | LC81240172019204LGN0 | 23 July 2019 | 124 | 17 |

**Table 2: Summary of the multiple linear regression for the environmental drivers of thaw depth. *SE* is the standard error of the corresponding variable's coefficient. The $R^2$ column displays the coefficient of determination of each predictor variable separately as well as for the multiple linear regression model.**

| Independent variables | Coefficient | *SE* | *T* | *p > \|t\|* | $R^2$ |
|---|---|---|---|---|---|
| **Site moisture** | -11.42 | 5.38 | -2.12 | 0.05 | 0.28 |
| **Vegetation density** | -5.51 | 2.80 | -1.97 | 0.07 | 0.20 |
| **Larch proportion** | -21.97 | 12.74 | -1.72 | 0.10 | 0.19 |
| **Geometrically Structured Composite Burn Index** | 15.04 | 3.47 | 4.33 | <0.01 | 0.20 |
| **Constant** | 158.45 | 14.00 | 11.32 | <0.01 | |
| **Multiple linear regression model** | | | | | 0.73 |

**Table 3: Summary of the multiple linear regression for the remote sensing proxies of thaw depth. *SE* is the standard error of the corresponding variable's coefficient. The $R^2$ column displays the coefficient of determination of each predictor variable separately as well as for the multiple linear regression model.**

| Independent variables | Coefficient | *SE* | *t* | *p > \|t\|* | $R^2$ |
|---|---|---|---|---|---|
| **Albedo** | 2850.85 | 930.55 | 3.06 | <0.01 | 0.03 |
| **differenced Normalized Burn Ratio** | 171.34 | 60.23 | 2.85 | 0.01 | 0.17 |
| **Land surface temperature** | -0.21 | 2.98 | -0.07 | 0.95 | 0.43 |
| **Pre-fire Normalized Difference Vegetation Index** | -406.38 | 143.49 | -2.83 | 0.01 | 0.10 |
| **Constant** | 163.36 | 919.10 | 0.18 | 0.86 | |
| **Multiple linear regression model** | | | | | 0.66 |



**Table 4: Comparison of environmental variables and remote sensing proxies of the adjacent burned-unburned plot pairs.**

| Plot identification label | ER019B | ER020C | ER021B | ER022C |
|---|---|---|---|---|
| **Burn or unburned** | Burned | Unburned | Burned | Unburned |
| **Topographic position** | Upland | Upland | Midslope | Midslope |
| **Site moisture** | Subxeric | Subxeric | Subxeric to Mesic | Subxeric to Mesic |
| **Elevation (m)** | 229.74 | 228.92 | 224.10 | 221.16 |
| **Stand age (year)** | 120 | 139 | 90 | 162 |
| **Basal area ($m^2.ha^{-1}$)** | 16.75 | 28.04 | 37.52 | 11.23 |
| **Vegetation density ($stems.m^{-2}$)** | 0.62 | 0.50 | 0.12 | 0.28 |
| **Larch proportion** | 0.46 | 0.93 | 1.00 | 0.88 |
| **Geometrically Structured Composite Burn Index** | 2.50 | | 2.56 | |
| **Burn depth (cm)** | 9.89 | | 10.52 | |
| **Land surface temperature (K)** | 309.25 | 304.14 | 306.51 | 303.62 |
| **Albedo** | 0.09 | 0.11 | 0.10 | 0.11 |
| **differenced Normalized Burn Ratio** | 0.66 | 0.01 | 0.37 | 0.00 |
| **Pre-fire Normalized Difference Vegetation Index** | 0.80 | 0.79 | 0.78 | 0.78 |
| **Thaw depth (cm)** | 150 | 97 | 172 | 99 |