# Peer review of "Environmental drivers and remote sensing proxies of post-fire thaw depth in Eastern Siberian larch forests"

_EGUsphere, 2024_

## Author Comment (AC1)

**This paper investigated post-fire thaw depth and its driving factors in East Siberian employing both field measurement and remote sensing proxies. Their findings indicate that: 1) fire exacerbates thaw depth, particularly when compared to unburned regions; 2) a combination of site moisture, forest composition, and fire severity accounts for 73.4% of thaw depth variability based on field investigations, while remote sensing proxies such as albedo, differenced Normalized Burn Ratio, land surface temperature, and NDVI contribute to explaining 66.3% of the variability.**

**The research explored topographical, vegetative, and burning effects on post-fire thaw depth in permafrost soil and mapped thaw depth with remote sensing data. Although the framework looks promising, some clarifications and elucidations are necessary to bolster more convincing findings. There are some limitations that I believe require further investigation.**

*We thank the reviewer for the constructive and valuable assessment of our paper. A point-by-point response is provided below. The original reviewer comments are in* **bold**, *author comments in italic, and manuscript amendments are given in* green.

**My major comments are:**

1. **The manual selection process of driving factors for the MLR model appears insufficiently rigorous. Despite burn depth showing the highest correlation with thaw depth, it's omitted from the regression model. Do you have any thoughts on this selection?**

*We thank the reviewer for pointing this out and we agree that the manual selection of the variables may have been somewhat subjective. Following your comment, we tested an automatic selection using a stepwise regression and we plan to include this revised regression in the revised manuscript.*

*We will adjust the '2.4 Statistical approach' section as follows:*

**2.4 Statistical approach**

Our statistical analysis consisted of two major components. First, we assessed the environmental drivers of post-fire thaw depth using our field observations. Second, we evaluated how remote sensing variables can be used to predict field-measured thaw depth. For both analyses, we used an ordinary least squares (OLS) multiple linear regression (MLR) model with thaw depth as the response variable using the "statsmodels" Python module (Seabold and Perktold, 2010). We hypothesized that post-fire thaw depth relates to variables associated with topographic position (topographic position, site moisture, slope, and elevation), vegetation (stand age, basal area, vegetation density, and larch proportion), and fire severity (GeoCBI and burn depth). The selection of the environmental variables for the MLR model was based on a stepwise regression approach testing both forward selection and backward elimination. We calculated the relative importance of the retained variables to the MLR model explanatory power using the "pingouin" Python module (Vallat, 2018), which is based on the R package "relaimpo" (Grömping, 2006). The $R^2$ from the MLR model is partitioned by averaging over orderings among regressors using simple unweighted averages as proposed by Lindeman et al. (1980), providing a decomposition of the explained variance into non-negative contributions. Finally, we developed a MLR model using the remote sensing metrics (post-fire albedo, dNBR, LST, and pre-fire NDVI) as potential predictor variables to assess their capacity as proxies for post-fire thaw depth. The resultant obtained MLR was then spatially extrapolated to derive a continuous map of post-fire thaw depth over the study area.

In addition to these statistical analyses, which included all field plots, we also focused separately on two burned-unburned plot pairs. While our study area captured a large variability in landscape position, vegetation characteristics, and fire severity, these plots had the advantage of being spatially adjacent with a road acting as the fire barrier. Because of this, the landscape and

vegetation characteristics of these plots were very similar. As a result, these plot pairs represent interesting study cases in which the fire effect on thaw depth can be unequivocally separated from the landscape and vegetation influences.

*The selection of the environmental variables based on the stepwise regression approach testing both forward selection and backward elimination resulted in the selection of the following variables: basal area, vegetation density and burn depth. Please find below the proposed new '3 Results' section:*

**3 Results**

**3.1 Environmental drivers of post-fire thaw depth**

On average, summer thaw was deeper in burned (mean = 127.3 cm, standard deviation (sd) = 27.7 cm) than in unburned (98.1 cm, sd = 26.9 cm) plots (Fig. R1). An independent t-test indicated that this difference is statistically significant ($p = 0.04$). Thaw was deeper in uplands (Figs. R2, R3a, and R3d) and well-drained (Figs. R2 and R3b) landscape positions; in open (Figs. R2 and R4b), mature (R2 and R3e), and mixed forests which included tree species other than Cajander larch (Figs. R2 and R3f); and in high severity burns (Figs. R2, R3g, and R4c).

Based on the results of the stepwise regression, both forward selection and backward elimination, we retained basal area, vegetation density, and burn depth as predictor variables for the MLR (Fig. R4). The multiple linear regression results indicated that the selected environmental drivers together explained 73.3 % of the thaw depth variability. Burn depth was the main contributor for the MLR model explanatory power (Table R1).

**3.2 Remote sensing proxies of thaw depth**

Within the fire perimeter, there was considerable spatial variability in albedo, dNBR, LST, and pre-fire NDVI (Figs. R5 and R6). Albedo values inside the fire perimeter had a mean of 0.10 (sd = 0.02), compared to mean of 0.12 (sd = 0.02) outside the perimeter (Figs. R5a and R6a). The dNBR values from our 13 burned field plots ranged between 0.34 and 0.85, with a mean of 0.57, while for the entire fire perimeter the mean was 0.41 (sd = 0.27, Fig. R5b and R6b). In addition, LST values inside the fire scar were on average 5.24 K higher than the LST values outside the fire scar (Figs. R5c and R6c). However, for some parts of the fire scar, LST values exceeded the surrounding unburned pixels by approximately 20 K. For the pre-fire NDVI, the distributions of values inside and outside the fire perimeter were similar (Figs. R5d and R6d) with a mean NDVI value of 0.78 (sd = 0.08) inside the perimeter and and a mean NDVI value of 0.79 (sd = 0.07) outside the perimeter.

Thaw tended to be deeper in the plots with lower summer albedo (Fig. R7a), higher dNBR, (Fig. R7b), higher LST (Fig. R7c) and lower pre-fire NDVI (Fig. R7d). The correlation with LST was the strongest, while the correlations between thaw depth and dNBR, albedo and pre-fire NDVI were not statistically significant at $p < 0.05$ (Fig. R8).

Stepwise forward selection indicated that LST was the only variable to be selected for the linear regression, capturing 42.9 % of the variability in field-measured thaw depth. By contrast, stepwise backward elimination pointed dNBR, albedo, and pre-fire NDVI as the variables to be retained in the MLR model. Taken together, these three remote sensing proxies explained 66.3% of field-measured thaw depth variability. dNBR was the most important predictor in this model configuration (Table R2).

The MLR model for thaw depth based on dNBR, albedo, and pre-fire NDVI enabled spatially continuous thaw depth estimates over the study area, including uncertainty estimates (Fig. R9). We chose this MRL configuration due to the higher $R^2$, when compared to LST alone. Post-fire thaw depth varied largely within the fire scar, for example between 83.49 cm (5th percentile) and 226.42 cm (95th percentile), with a mean of 139.72 cm (sd = 43.76 cm). In contrast, for the pixels outside the fire perimeter, the estimated thaw depth varied between 66.59 cm (5th percentile) and 237.99 cm (95th percentile), with a mean of 125.08 cm

(sd = 52.15 cm) (Fig. R9c). On average, burned areas thus experienced 14.64 cm deeper summer thaw depth one year after the fire than unburned areas.

2. **Could you provide the significance for all the correlation matrices in Figures A1 and B1? Burn depth exhibits a positive correlation to thaw depth (0.53) and soil moisture (0.04), respectively, however, the thaw depth has a negative correlation (-0.53) to site moisture. This raises questions.**

*We plan to add the significance of the correlations in the matrices in new figure versions of the revised manuscript. Below you can find these revised figures (Figs. R2 and R8).*

*Site moisture is a plot-level site moisture classification* (Johnstone et al., 2008)*, defined as the "potential moisture available for plant growth". This classification assesses the site moisture based on local topographic drainage and also accounts for permafrost presence and soil texture. It results in a six-point scale (xeric, subxeric, subxeric to mesic, mesic, mesic to subhygric, and subhygric) ranging between dry and wet. As site moisture is a categorical variable, we ranked it on ordinal scale to include this variable in our linear analysis. Thus, when thaw depth shows a negative correlation with site moisture, that is indicative that drier (well-drained) plots tended to have deeper thaw. We will clarify this information in revision.*

**2.2 Field data**

[...]

We also assigned the topographic position of the plot in relation to its surroundings (upland, midslope, lowland) and plot-level site moisture classes following Johnstone et al. (2008). This approach assesses site moisture based on local topographic drainage thereby accounting for soil texture and permafrost presence. The resulting ordinal scale consists of six site moisture classes ranging between dry and wet (xeric, subxeric, subxeric to mesic, mesic, mesic to subhygric, and subhygric). The six classes represent the potential moisture availability for plant growth and should not be confused with temporally explicit soil moisture measurements. This site moisture classification has been used extensively in fire studies in boreal North America (Walker et al., 2020).

**4 Discussion**

**4.1 Environmental drivers of post-fire thaw depth**

[...]

Increasing fire severity tends to increase thaw depth (Alexander et al., 2018; Holloway et al., 2020; Jafarov et al., 2013; Jiang et al., 2015; Li et al., 2019). The positive correlations between the fire severity proxies burn depth and GeoCBI, and thaw depth found in our study is consistent with these previous results. In addition, fire severity is often influenced by vegetation and topographic conditions. In our study for example, the well-drained upland plots were characterized by deeper thaw , while lowland areas are usually wetter with limited drainage and shallow thaw (Figs. R2 and R3). These wet lowland ecosystems also tend to burn with lower severity  (Benscoter et al., 2011; Dillon et al., 2011; Holloway et al., 2020; Turetsky et al., 2011).

3. **Regarding the application of multi-linear regression, are you utilizing the original data or standardized data? Expanding on this in section 2.4 Statistical Approach would enhance clarity.**

*We used the original data. We will expand the statistical approach to include this information as suggested. A proposed new version of the '2.4 Statistical approach' section can be found in the answer to comment 1.*

4. **Are the environmental factors and remote sensing proxies of thaw depth consistent between burned and unburned plots if you explore the data separately? How does the correlation coefficient fluctuate between burned and unburned regions?**

*We acknowledge that our field dataset is somewhat limited in size. As a result, we feel that we cannot confidently infer any statistically relevant information by analyzing only 6 unburned plots. However, we would like to point out that the variables related to landscape position do not change in their definition between burned and unburned sites. The vegetation characteristics were based on a reconstruction of the pre-fire situation. Finally, for the fire severity measurements (namely burn depth and GeoCBI, where the unburned plots receive the values of 0) the correlation information when considering only the burned plots is shown in Fig. R10. When considering only burned plots, burn depth, the fire severity variable retained in the revised model, has a correlation coefficient of 0.55 with thaw depth, compared to 0.53 when all plots were analyzed.*

5. **According to the MLR model, site moisture seems to play a more significant role in driving variations in thaw depth than fire severity. However, thaw depth in burned areas typically surpasses that of unburned areas on average. How do you consider the relative contributing importance of site moisture and burning severity? What are your thoughts on the potential driving mechanism of thaw depth by a comprehensive interpretation of the statistical model in this study? Furthermore, given the potential contribution of site moisture to thaw depth, why wasn't soil moisture remote sensing data considered?**

*First, as showed in our answer to comment 1, with the revised stepwise selection of variables, site moisture is no longer retained in the MLR model. We appreciate the good suggestion of including the relative importance of the variables in the model. The inclusion of this metric is presented in the answer to comment 1. We have also discussed the relation between site moisture and fire severity in our answer to comment 2*

**Other small comments:**

6. **Line 118: The reference for Johnstone et al. (2008) is missing.**

*We thank the reviewer for the comment. However, we have checked and the reference is there in the list (line 528, in the first version of the manuscript).*

*Johnstone, J. F., Hollingsworth, T. N., and Chapin, F. S.: A key for predicting postfire successional trajectories in black spruce stands of interior Alaska., https://doi.org/10.2737/PNW-GTR-767, 2008.*

7. **Line 234: Does the larch tree play a certain function in inducing boreal fires? What is the reason for retaining the larch proportion?**
8. **Lines 280 – 282: why do plots with fewer larch trees thaw deeper? You may expand some discussion here on larch proportion.**

*Thanks for the comments, after the new selection of field variables, larch proportion is no longer retained in the MLR model. Regardless, we will expand this discussion about the role of larch trees on permafrost dynamics in the revised version of the manuscript.*

Moreover, the presence of Cajander larch trees is closely coupled with permafrost (Herzschuh, 2020), as they maintain permafrost by controlling its seasonal thawing. In turn, the permafrost helps to provide sufficient water to the trees by preventing it from draining away quickly (Zhang et al., 2011). Our results reflect this vegetation-permafrost interaction, where plots with fewer larch trees showed a more pronounced thaw. Our observations in the field also revealed that larch trees were prevalent and tend to dominate in mesic and hygric parts of the landscape, including the lowlands with shallower permafrost thaw. Conversely, Scots pine was more prevalent in the drier upland areas, which showed deeper thaw (Fig. R2). Scots pine trees in the region are found on moderately warm and dry locations, for example on hilltops and well-drained summits of watersheds with sandy soils and deeper permafrost thaw. In contrast, larch trees are dominant in wide depressions with often cool waterlogged soils with shallower permafrost thaw (Isaev et al., 2010). Eastern Siberian larch trees are some of the only

tree species that can successfully grow on permafrost soils with very shallow thaw. This is due to their ability to develop the adventitious rooting system (Herzschuh, 2020; Kajimoto, 2010). Larch and also Scots pine have evolved under periodic fire conditions, with the capacity to regenerate and grow after fire (Kharuk et al., 2021). The competitive advantages have led to the establishment of larch dominance in Siberian permafrost environments. However, increasing permafrost degradation could lead to a shift in the dominant species, with larch being replaced by pine and other species (Zhang et al., 2011).

9. **Figures A1 & B1: please add significance to the correlation matrix.**

*We will add significance to the correlation matrices as suggested (Figs. R2 and R8).*

10. **Figures A2 & 3: For unburned areas where burn depth and GeoCBI are 0, it's worth showing what drives the thaw depth when there is no fire rather than explaining everything by one statistic model.**

*Thanks for the comments. We have already addressed them in our response to comment 4.*

11. **Line 405: please double-check all the references.**

*Thanks for pointing this out, we have double-checked all references.*

12. **Figure 4 (b) & (d): The fire scar wasn't fully covered.**
13. **Figure 8 (a) & (b): The fire scar wasn't fully covered.**

*Unfortunately, there is no pre-fire image available that covers the northwestern tip of the fire scar. Figure 4b and d (in the first version of the manuscript) show that there is no data for that small portion. We will clarify this further in the caption (Fig. R5).*

*References*

[revised manuscript text omitted]

**Figure R10. Correlation matrix between field-measured environmental variables representing landscape, vegetation and fire severity characteristics, and thaw depth, including only burned plots.**

**Table R1: Summary of the multiple linear regression for the environmental drivers of thaw depth.** *SE* **is the standard error of the corresponding variable's coefficient. The R² column displays the coefficient of determination of each predictor variable separately as well as for the multiple linear regression model. The relative importance indicates the contribution of each independent variable to the multiple linear regression model R².**

| Independent variables | Coefficient | *SE* | *t* | *p > |t|* | R² | Relative importance (%) |
|---|---|---|---|---|---|---|
| Basal area | 1.51 | 0.51 | 2.96 | <0.001 | 0.07 | 14.7 |
| Vegetation density | -10.47 | 2.24 | -4.68 | 0.009 | 0.20 | 38.4 |
| Burn depth | 3.88 | 0.78 | 4.96 | <0.001 | 0.28 | 46.9 |
| Constant | 82.36 | 13.32 | 6.18 | <0.001 | | |
| Multiple linear regression model | | | | | 0.73 | |

**Table R2: Summary of the multiple linear regression for the remote sensing proxies of thaw depth.** *SE* is the standard error of the corresponding variable's coefficient. The $R^2$ column displays the coefficient of determination of each predictor variable separately as well as for the multiple linear regression model. The relative importance indicates the contribution of each independent variable to the multiple linear regression model $R^2$.

| Independent variables | Coefficient | *SE* | *t* | *p > \|t\|* | $R^2$ | Relative importance (%) |
|---|---|---|---|---|---|---|
| Albedo | 2807.88 | 674.09 | 4.16 | 0.001 | 0.03 | 24.0 |
| differenced Normalized Burn Ratio | 167.89 | 33.01 | 5.08 | <0.001 | 0.17 | 47.7 |
| Pre-fire Normalized Difference Vegetation Index | -399.24 | 97.12 | -4.11 | 0.001 | 0.10 | 28.3 |
| Constant | 99.64 | 77.27 | 1.29 | 0.216 | | |
| Multiple linear regression model | | | | | 0.66 | |

---

## Author Comment (AC2)

In this manuscript, the authors investigated post-fire thaw depth within one fire event in the Republic of Sakha. They used a combination of field data collected one year post-burn and compare this field data with multiple remote sensing indices derived from Landsat optical and thermal data. The environmental characteristics assessed included a variety of vegetation, fire severity, and thaw depth characteristics. The remote sensing techniques included several pre- and post-fire indices, including land surface temperature. Through their field work, the authors found deeper thaw in burned areas and well-drained areas. The authors found that the remote sensing characteristics assessed explained 66.3% of the variability in the field-measured thaw depth. Additionally, it was found that land surface temperature correlated highly with post-fire thaw depth (42.9% of the variability explained).

This was a well-written manuscript which clearly described the research planned and conducted, both in the field, and with the remote sensing techniques. The use of Landsat thermal data to assess thaw depth was a new application, and it was surprising that the correlation was so high, especially considering that the resolution of the data was 100m. The discussion section mentioned some of the concerns with these new techniques and adequately addressed them, including the resolution of the Landsat thermal data, the small sample size of the field dataset, and the timing of the collection of the field data (mid-summer, as opposed to end of summer when active layer thickness could be collected). The authors also provided a worthwhile discussion of future research including the use of more advanced machine learning techniques, collecting additional field data, and incorporating radar data into such an analysis in the future.

*We thank the reviewer for the constructive and valuable assessment of our paper. A point-by-point response is provided below. The original reviewer comments are in* **bold**, *author comments in italic, and manuscript amendments are given in* green.

**Comments:**

1. **Line 22 and 232 – Was the thaw depth significantly deeper in burned than unburned plots? The mean and standard deviation are provided, but the significance level is not. Please provide it if possible.**

*Thank you for the comment. We will address this in the revised version of the manuscript:*

On average, summer thaw was deeper in burned (mean = 127.3 cm, standard deviation (sd) = 27.7 cm) than in unburned (98.1 cm, sd = 26.9 cm) plots (Fig. 2). An independent t-test indicated that this difference is statistically significant ($p$ = 0.04).

2. **Section 2.3 – Consider a table to show the indices used and the formulas, as a way for readers to have a quick overview. Perhaps this could go in the Appendix.**

*Thank you for your suggestion. We will add a table to the Appendix as suggested (Table R1).*

3. **Line 160-162 – The pre-fire imagery is from 2 years prior to the fire, and 2 scenes needed to be mosaicked together to cover the entire fire event – Could this have affected any of the results? Consider adding a clarifying statement in either the methods or discussion section.**

*We understand the reviewer's concern. However, we do not expect that using a pre-fire image from two years before the fire event significantly impacted our results, because it was representative of the environmental conditions before the fire, since no other disturbance occurred in the area between two years and one year before the fire. This is in line with recommendations from Key and Benson (2006) who stated that the acquisition of pre-fire imagery can safely be from two to three years before the fire, as long as other landscape disturbances do not interfere with the subject burn. We will add a statement to the methods as suggested:*

For the pre-fire imagery, we used a cloud-free image from July 7, 2016, near the anniversary date of the post-fire imagery. No cloud-free summer images were available from 2017. This timing aligns with recommendations from Key and Benson (2006), since no other disturbance had occurred between two years and one year before the fire.

Furthermore, we indeed mosaicked two Landsat scenes to acquire near-full coverage of the fire scar. Both before and after the fire, these two scenes were from the same day. As a result, the mosaicked scenes represent comparable environmental conditions.

4. **Line 289 – The case studies of the 2 burned/unburned plot pairs undoubtedly helped in separating the impact of fire on thaw from topographic and vegetation influences, but this is still a very small sample size, and should be treated as such. Perhaps soften the language here from "enabled", to show that this small sample size would not fully address all situations in separating influences on thaw depth.**

We will use a softer term as suggested by the review. Please find the proposed new sentence below:

Our case studies of two burned-unburned plot pairs divided by a fire barrier, though limited in sample size, helped us to separate the impact of fire on the active layer thaw from topographic and vegetation influences.

5. **Figure A2 – Figure 2 provides a description of the meaning of the triangle and the bounds of the box plot, but it is not repeated in Figure A2. Consider adding it again here as the Appendix is separate from the main manuscript and readers could be confused.**

Thank you for pointing this out. We will add the description to the figure as suggested. Please see Figure R1.

References

Key, C. H. and Benson, N. C.: Landscape Assessment (LA), in: FIREMON: Fire effects monitoring and inventory system, edited by: Lutes, D. C. ;, Keane, R. E. ;, Caratti, J. F. ;, Key, C. H. ;, Benson, N. C. ;, Sutherland, S., and Gangi, L. J., U.S. Department of Agriculture, Forest Service, Rocky Mountain Research Station., Fort Collins, CO, 1–55, 2006.

[Figure]

**Figure R1: Relationships between (a) topographic position, (b) site moisture, (c) slope, (d) elevation, (e) stand age, (f) larch proportion, (g) Geometrically structured Composite Burn Index, and thaw depth. In (a) and (b), each box ranges from the first to the third quartile. Whiskers extend to points that lie within 1.5 times the interquartile range. The median is indicated by the horizontal line and the mean by the black triangle. For site moisture, classes were grouped together for better visualization.**

**Table R1: Summary of the remote sensing metrics used in the study. $\rho$ represents the reflectance of the Landsat 8 Operational Land Imager bands. $L_\lambda^{sen}$ is the Landsat 8 Thermal Infrared Sensor band 10 radiance. $K_1$ and $K_2$ are calibration constants. $L_\lambda^{\downarrow}$ and $L_\lambda^{\uparrow}$ are the downwelling and upwelling atmospheric radiances and $\tau_\lambda$ is the atmospheric transmittance.**

| Remote sensing metric | Equation |
|---|---|
| Normalized Difference Vegetation Index (NDVI) | $NDVI = \dfrac{\rho_5 - \rho_4}{\rho_5 + \rho_4}$ |
| Albedo ($\alpha$) | $\alpha = 0.356\rho_2 + 0.130\rho_4 + 0.373\rho_5 + 0.085\rho_6 + 0.072\rho_7 - 0.0018$ |
| differenced Normalized Burn Ratio (dNBR) | $dNBR = NBR_{pre-fire} - NBR_{post-fire}$ , where $NBR = \dfrac{\rho_5 - \rho_7}{\rho_5 + \rho_7}$ |
| Fractional vegetation cover ($P_V$) | $P_V = \left(\dfrac{NDVI - NDVI_{min}}{NDVI_{max} - NDVI_{min}}\right)^2$, where $NDVI_{min} = 0.2$ and $NDVI_{max} = 0.5$ |
| Land surface emissivity ($\varepsilon$) | $\varepsilon = \begin{cases} 0.962 & 0.0 \leq NDVI < 0.2 \\ 0.990P_V + 0.962(1 - P_V) + d\varepsilon & 0.2 \leq NDVI \leq 0.5 \\ 0.990 & NDVI > 0.5 \\ 0.993 & NDVI < 0.0 \end{cases}$

 $d\varepsilon = (1 - 0.962)0.990F'(1 - P_V)$ , where $F' = 0.55$ |

Land surface temperature (LST)

$$LST = \frac{K_2}{ln\left(\frac{K_1}{\frac{L_\lambda^{sen} - L_\lambda^\uparrow - \tau_\lambda(1 - \varepsilon_\lambda)L_\lambda^\downarrow}{\tau_\lambda \varepsilon_\lambda}} + 1\right)}$$

---

## Author Response (AR1)

*Dear Editor and Reviewers,*

*Thank you for considering our manuscript for publication in ESD. We appreciate your time and your valuable comments.*

*A response to the reviews is provided below. The original reviewer comments are in **bold**, author comments in italic, and manuscript amendments are given in green.*

**Referee #1**

**This paper investigated post-fire thaw depth and its driving factors in East Siberian employing both field measurement and remote sensing proxies. Their findings indicate that: 1) fire exacerbates thaw depth, particularly when compared to unburned regions; 2) a combination of site moisture, forest composition, and fire severity accounts for 73.4% of thaw depth variability based on field investigations, while remote sensing proxies such as albedo, differenced Normalized Burn Ratio, land surface temperature, and NDVI contribute to explaining 66.3% of the variability.**

**The research explored topographical, vegetative, and burning effects on post-fire thaw depth in permafrost soil and mapped thaw depth with remote sensing data. Although the framework looks promising, some clarifications and elucidations are necessary to bolster more convincing findings. There are some limitations that I believe require further investigation.**

*We thank the reviewer for the constructive and valuable assessment of our paper. A point-by-point response is provided below. The original reviewer comments are in **bold**, author comments in italic, and manuscript amendments are given in green.*

**My major comments are:**

1. **The manual selection process of driving factors for the MLR model appears insufficiently rigorous. Despite burn depth showing the highest correlation with thaw depth, it's omitted from the regression model. Do you have any thoughts on this selection?**

*We thank the reviewer for pointing this out and we agree that the manual selection of the variables may have been somewhat subjective. Following your comment, we have performed an automatic selection using a stepwise regression and have included this revised regression in the revised manuscript.*

*We have adjusted the '2.4 Statistical approach' section as follows (lines 215-236 in the revised version of the manuscript):*

**2.4 Statistical approach**

Our statistical analysis consisted of two major components. First, we assessed the environmental drivers of post-fire thaw depth using our field observations. Second, we evaluated how remote sensing variables can be used to predict field-measured thaw depth. For both analyses, we used an ordinary least squares (OLS) multiple linear regression (MLR) model with thaw depth as the response variable using the "statsmodels" Python module (Seabold and Perktold, 2010). We hypothesized that post-fire thaw depth relates to variables associated with topography (topographic position, site moisture, slope, and elevation), vegetation (stand age, basal area, vegetation density, and larch proportion), and fire severity (GeoCBI and burn depth). We used the original data (i.e., non-standardized) and ranked the categorical variables topographic position and site moisture on ordinal scale to include them in the linear analysis. The selection of the environmental variables for the MLR model was based on a stepwise regression approach testing both forward selection and backward elimination. We calculated the relative

[revised manuscript text omitted]

2. **Could you provide the significance for all the correlation matrices in Figures A1 and B1? Burn depth exhibits a positive correlation to thaw depth (0.53) and soil moisture (0.04), respectively, however, the thaw depth has a negative correlation (-0.53) to site moisture. This raises questions.**

*We have added the significance of the correlations in the matrices in new figure versions of the revised manuscript. Below you can find these revised figures (Figs. 3 and C1).*

*Site moisture is a plot-level site moisture classification* (Johnstone et al., 2008)*, defined as the "potential moisture available for plant growth". This classification assesses the site moisture based on local topographic drainage and also accounts for permafrost presence and soil texture. It results in a six-point scale (xeric, subxeric, subxeric to mesic, mesic, mesic to subhygric, and subhygric) ranging between dry and wet. As site moisture is a categorical variable, we ranked it on ordinal scale to include this variable in our linear analysis. Thus, when thaw depth shows a negative correlation with site moisture, that is indicative that drier (well-drained) plots tended to have deeper thaw. We have clarified this information in the revision (lines 117-124 and lines 313-318).*

**2.2 Field data**

[...]

We also assigned the topographic position of the plot in relation to its surroundings (upland, midslope, lowland) and plot-level site moisture classes following Johnstone et al. (2008). This approach assesses site moisture based on local topographic drainage thereby accounting for soil texture and permafrost presence. The resulting ordinal scale consists of six site moisture classes ranging between dry and wet (xeric, subxeric, subxeric to mesic, mesic, mesic to subhygric, and subhygric). The six classes represent the potential moisture available for plant growth and should not be confused with actual measurements of soil moisture. This site moisture classification has been used extensively for fire studies in boreal North America (Dieleman et al., 2020; Walker et al., 2018a, b, 2020).

**4 Discussion**

**4.1 Environmental drivers of post-fire thaw depth**

[...]

Increasing fire severity tends to increase thaw depth (Alexander et al., 2018; Holloway et al., 2020; Jafarov et al., 2013; Jiang et al., 2015; Li et al., 2019). The positive correlations between burn depth and GeoCBI, and thaw depth found in this study are consistent with these previous results. In addition, fire severity is often influenced by vegetation and topographic conditions.

In our study for example, the well-drained plots are on upland landscape positions with deeper thaw depths, while lowland areas are usually wetter with limited drainage and shallow thaw (Figs. 3 and A2). These wet lowland ecosystems also tend to burn with lower severity (Benscoter et al., 2011; Dillon et al., 2011; Holloway et al., 2020; Turetsky et al., 2011).

3. **Regarding the application of multi-linear regression, are you utilizing the original data or standardized data? Expanding on this in section 2.4 Statistical Approach would enhance clarity.**

*We used the original data. We have expanded the statistical approach to include this information as suggested. A proposed new version of the '2.4 Statistical approach' section can be found in the answer to comment 1.*

4. **Are the environmental factors and remote sensing proxies of thaw depth consistent between burned and unburned plots if you explore the data separately? How does the correlation coefficient fluctuate between burned and unburned regions?**

*We acknowledge that our field dataset is somewhat limited in size. As a result, we feel that we cannot confidently infer any statistically relevant information by analyzing only 7 unburned plots. However, we would like to point out that the variables related to landscape position do not change in their definition between burned and unburned sites. The vegetation characteristics were based on a reconstruction of the pre-fire situation. Finally, for the fire severity measurements (namely burn depth and GeoCBI, where the unburned plots receive the values of 0) the correlation information when considering only the burned plots is shown in Fig. R1. When considering only burned plots, burn depth, the fire severity variable retained in the revised model, has a correlation coefficient of 0.55 with thaw depth, compared to 0.53 when all plots were analyzed.*

5. **According to the MLR model, site moisture seems to play a more significant role in driving variations in thaw depth than fire severity. However, thaw depth in burned areas typically surpasses that of unburned areas on average. How do you consider the relative contributing importance of site moisture and burning severity? What are your thoughts on the potential driving mechanism of thaw depth by a comprehensive interpretation of the statistical model in this study? Furthermore, given the potential contribution of site moisture to thaw depth, why wasn't soil moisture remote sensing data considered?**

*First, as shown in our answer to comment 1, with the revised stepwise selection of variables, site moisture is no longer retained in the MLR model. We appreciate the good suggestion of including the relative importance of the variables in the model. The inclusion of this metric is presented in the answer to comment 1. We have also discussed the relation between site moisture and fire severity in our answer to comment 2*

**Other small comments:**

6. **Line 118: The reference for Johnstone et al. (2008) is missing.**

*We thank the reviewer for the comment. However, we have checked and the reference is there in the list (line 528 in the first version of the manuscript and line 628 in the revised version).*

*Johnstone, J. F., Hollingsworth, T. N., and Chapin, F. S.: A key for predicting postfire successional trajectories in black spruce stands of interior Alaska., https://doi.org/10.2737/PNW-GTR-767, 2008.*

7. **Line 234: Does the larch tree play a certain function in inducing boreal fires? What is the reason for retaining the larch proportion?**
8. **Lines 280 – 282: why do plots with fewer larch trees thaw deeper? You may expand some discussion here on larch proportion.**

*Thanks for the comments, after the new selection of field variables, larch proportion is no longer retained in the MLR model. Regardless, we have expanded this discussion about the role of larch trees on permafrost dynamics in the revised version of the manuscript (lines 291-305).*

Moreover, the presence of Cajander larch trees is closely coupled with permafrost (Herzschuh, 2020). In turn, permafrost helps to provide sufficient water to the trees by preventing it from draining away quickly (Zhang et al., 2011). Our results reflect this vegetation-permafrost interaction, where plots with fewer larch trees showed more pronounced thaw. Our field observations also showed that larch trees were prevalent and tend to dominate in mesic and hydric parts of the landscape, including in lowlands with shallow permafrost thaw. Conversely, Scots pine was more prevalent in the drier upland areas, which demonstrated deeper thaw (Fig. 3). Scots pine trees in our study region tend to occur on moderately warm and dry landscape positions such as ridges and hilltops. These locations were characterized by sandy soils and deeper thaw. In contrast, larch trees are dominant in lowland depressions of watersheds, characterized by cool and often waterlogged soils and shallower permafrost thaw (Isaev et al., 2010). Eastern Siberian larch trees are one of the only tree species that can successfully grow on permafrost soils with very shallow thaw depth. This is due to their ability to develop an adventitious rooting system (Herzschuh, 2020; Kajimoto, 2010). Larch and also Scots pine have evolved under periodic fire conditions, with the capacity to regenerate and grow after fire (Kharuk et al., 2021). These competitive advantages have led to the establishment of larch dominance in Siberian permafrost environments, and fire occurrence has been a prerequisite for this evolutionary process. However, increased permafrost degradation could lead to a shift in dominant tree species, with larch trees being replaced by pine trees and other tree species (Zhang et al., 2011).

9. **Figures A1 & B1: please add significance to the correlation matrix.**

*We have added significance to the correlation matrices as suggested (Figs. 3 and C1, lines 993 and 417).*

10. **Figures A2 & 3: For unburned areas where burn depth and GeoCBI are 0, it's worth showing what drives the thaw depth when there is no fire rather than explaining everything by one statistic model.**

*Thanks for the comments. We have already addressed them in our response to comment 4.*

11. **Line 405: please double-check all the references.**

*Thanks for pointing this out, we have double-checked all references.*

12. **Figure 4 (b) & (d): The fire scar wasn't fully covered.**
13. **Figure 8 (a) & (b): The fire scar wasn't fully covered.**

*Unfortunately, there is no pre-fire image available that covers the northwestern tip of the fire scar. Figure 4b and d (in the first version of the manuscript) show that there is no data for that small portion. We have clarified this further in the captions (Figs. 5 and 8).*

*References*

[revised manuscript text omitted]

Referee #2

In this manuscript, the authors investigated post-fire thaw depth within one fire event in the Republic of Sakha. They used a combination of field data collected one year post-burn and compare this field data with multiple remote sensing indices derived from Landsat optical and thermal data. The environmental characteristics assessed included a variety of vegetation, fire severity, and thaw depth characteristics. The remote sensing techniques included several pre- and post-fire indices, including land surface temperature. Through their field work, the authors found deeper thaw in burned areas and well-drained areas. The authors found that the remote sensing characteristics assessed explained 66.3% of the variability in the field-measured thaw depth. Additionally, it was found that land surface temperature correlated highly with post-fire thaw depth (42.9% of the variability explained).

This was a well-written manuscript which clearly described the research planned and conducted, both in the field, and with the remote sensing techniques. The use of Landsat thermal data to assess thaw depth was a new application, and it was surprising that the correlation was so high, especially considering that the resolution of the data was 100m.  The discussion section mentioned some of the concerns with these new techniques and adequately addressed them, including the resolution of the Landsat thermal data, the small sample size of the field dataset, and the timing of the collection of the field data (mid-summer, as opposed to end of summer when active layer thickness could be collected). The authors also provided a worthwhile discussion of future research including the use of more advanced machine learning techniques, collecting additional field data, and incorporating radar data into such an analysis in the future.

*We thank the reviewer for the constructive and valuable assessment of our paper. A point-by-point response is provided below. The original reviewer comments are in* **bold**, *author comments in italic, and manuscript amendments are given in* green.

**Comments:**

1. **Line 22 and 232 – Was the thaw depth significantly deeper in burned than unburned plots? The mean and standard deviation are provided, but the significance level is not. Please provide it if possible.**

*Thank you for the comment. We have addressed this in the revised version of the manuscript (lines 239-240):*

On average, summer thaw depth was deeper in burned (mean = 127.3 cm, standard deviation (sd) = 27.7 cm) than in unburned (98.1 cm, sd = 26.9 cm) plots (Fig. 2). This difference was statistically significant at $p < 0.05$.

2. **Section 2.3 – Consider a table to show the indices used and the formulas, as a way for readers to have a quick overview. Perhaps this could go in the Appendix.**

*Thank you for your suggestion. We have added a table to the Appendix as suggested (Table A1, line 404).*

3. **Line 160-162 – The pre-fire imagery is from 2 years prior to the fire, and 2 scenes needed to be mosaicked together to cover the entire fire event – Could this have affected any of the results? Consider adding a clarifying statement in either the methods or discussion section.**

*We understand the reviewer's concern. However, we do not expect that using a pre-fire image from two years before the fire event significantly impacted our results, because it was representative of the environmental conditions before the fire, since no other disturbance occurred in the area between two years and one year before the fire. This is in line with recommendations from* Key and Benson (2006) *who stated that the acquisition of pre-fire imagery can safely be from two to three years before the fire, as long as no other landscape disturbances interfere with the subject fire. We have added a statement to the methods as suggested (lines 163-167):*

For the pre-fire imagery, we used a cloud-free image from July 7, 2016, near the anniversary date of the post-fire imagery. No cloud-free summer images were available from 2017. The acquisition timing of the pre-fire image aligns with recommendations of Key and Benson (2006), since no other disturbance in the landscape interfered with the subject fire between the summers of 2016 and 2017. Two scenes were mosaicked to cover the entire fire perimeter for both pre- and post-fire imagery.

*Furthermore, we indeed mosaicked two Landsat scenes to acquire near-full coverage of the fire scar. Both before and after the fire, these two scenes were from the same day. As a result, the mosaicked scenes represent comparable environmental conditions.*

4. **Line 289 – The case studies of the 2 burned/unburned plot pairs undoubtedly helped in separating the impact of fire on thaw from topographic and vegetation influences, but this is still a very small sample size, and should be treated as such. Perhaps soften the language here from "enabled", to show that this small sample size would not fully address all situations in separating influences on thaw depth.**

*We have used a softer term as suggested by the review. Please find the new sentence below (lines 311-312):*

Our case studies of two burned-unburned plot pairs divided by a fire barrier, though limited in sample size, helped us to separate the impact of fire on the active layer thaw from topographic and vegetation influences.

5. **Figure A2 – Figure 2 provides a description of the meaning of the triangle and the bounds of the box plot, but it is not repeated in Figure A2. Consider adding it again here as the Appendix is separate from the main manuscript and readers could be confused.**

*Thank you for pointing this out. We have added the description to the figure as suggested. Please see Figure B1 below (line 409 in the revised manuscript).*

*References*

*Key, C. H. and Benson, N. C.: Landscape Assessment (LA), in: FIREMON: Fire effects monitoring and inventory system, edited by: Lutes, D. C. ;, Keane, R. E. ;, Caratti, J. F. ;, Key, C. H. ;, Benson, N. C. ;, Sutherland, S., and Gangi, L. J., U.S. Department of Agriculture, Forest Service, Rocky Mountain Research Station., Fort Collins, CO, 1–55, 2006.*

**Table A1: Summary of the remote sensing metrics used in the study.** $\rho$ **represents the reflectance of the Landsat 8 Operational Land Imager (OLI) bands.** $L_\lambda^{sen}$ **refers to the Landsat 8 Thermal Infrared Sensor (TIRS) band 10 radiance.** $K_1$ **and** $K_2$ **are calibration constants.** $L_\lambda^\downarrow$ **and** $L_\lambda^\uparrow$ **are the downwelling and upwelling atmospheric radiances and** $\tau_\lambda$ **is the atmospheric transmittance.**

| Remote sensing metric | Equation |
|---|---|
| Normalized Difference Vegetation Index (NDVI) | $NDVI = \dfrac{\rho_5 - \rho_4}{\rho_5 + \rho_4}$ |
| Albedo ($\alpha$) | $\alpha = 0.356\rho_2 + 0.130\rho_4 + 0.373\rho_5 + 0.085\rho_6 + 0.072\rho_7 - 0.0018$ |
| differenced Normalized Burn Ratio (dNBR) | $dNBR = NBR_{pre-fire} - NBR_{post-fire}$ , where $NBR = \dfrac{\rho_5 - \rho_7}{\rho_5 + \rho_7}$ |
| Fractional vegetation cover ($P_V$) | $P_V = \left(\dfrac{NDVI - NDVI_{min}}{NDVI_{max} - NDVI_{min}}\right)^2$, where $NDVI_{min} = 0.2$ and $NDVI_{max} = 0.5$ |
| Land surface emissivity ($\varepsilon$) | $\varepsilon = \begin{cases} 0.962 & 0.0 \leq NDVI < 0.2 \\ 0.990 P_V + 0.962(1 - P_V) + d\varepsilon & 0.2 \leq NDVI \leq 0.5 \\ 0.990 & NDVI > 0.5 \\ 0.993 & NDVI < 0.0 \end{cases}$
 $d\varepsilon = (1 - 0.962)0.990 F'(1 - P_V)$ , where $F' = 0.55$ |
| Land surface temperature (LST) | $LST = \dfrac{K_2}{ln\left(\dfrac{K_1}{\dfrac{L_\lambda^{sen} - L_\lambda^\uparrow - \tau_\lambda(1 - \varepsilon_\lambda)L_\lambda^\downarrow}{\tau_\lambda \varepsilon_\lambda}} + 1\right)}$ |

[Figure]

**Figure B1: Relationships between (a) topographic position, (b) site moisture, (c) slope, (d) elevation, (e) stand age, (f) larch proportion, (g) Geometrically structured Composite Burn Index, and thaw depth. In (a) and (b), each box ranges from the first to the third quartile. Whiskers extend to points that lie within 1.5 times the interquartile range. The median is indicated by the horizontal line and the mean by the black triangle. For site moisture, classes were grouped together for better visualization. In the regression lines, shading indicates the 95% confidence interval and *r* is the Pearson correlation coefficient.**